# Scaling Laws of Global Weather Models

Yuejiang Yu [1]   Langwen Huang [1]   Alexandru Calotoiu [1]   Torsten Hoefler [1]

## Abstract

Data-driven models are revolutionizing weather forecasting. To optimize training efficiency and model performance, this paper analyzes empirical scaling laws within this domain. We investigate the relationship between model performance (validation loss) and three key factors: model size ($N$), dataset size ($D$), and compute budget ($C$). Across a range of models, we find that Aurora exhibits the strongest data-scaling behavior: increasing the training dataset by 10× reduces validation loss by up to 3.2×. GraphCast demonstrates the highest parameter efficiency, yet suffers from limited hardware utilization. Our compute-optimal analysis indicates that, under fixed compute budgets, allocating resources to more total training data yields greater performance gains than increasing model size. Furthermore, we analyze model shape and uncover scaling behaviors that differ fundamentally from those observed in language models: weather forecasting models consistently favor increased width over depth. These findings suggest that future weather models should prioritize wider architectures and larger effective training datasets to maximize predictive performance.

## 1. Introduction

Few scientific problems possess the immediate global impact and computational complexity of weather prediction. Traditional numerical weather prediction relies on physics-based models that simulate atmospheric dynamics through partial differential equations. With the emergence of data-driven forecasting approaches, machine learning-based methods have been growing rapidly (Lam et al., 2023; Bi et al., 2023; Bonev et al., 2023; Chen et al., 2023a;b; Bodnar et al., 2024; Lang et al., 2024) and revolutionized weather forecasting with their increasing competitiveness (Bauer

[1]Department of Computer Science, ETH Zurich, Zurich, Switzerland. Correspondence to: Torsten Hoefler <torsten.hoefler@inf.ethz.ch>.

*Proceedings of the 43$^{rd}$ International Conference on Machine Learning*, Seoul, South Korea. PMLR 306, 2026. Copyright 2026 by the author(s).

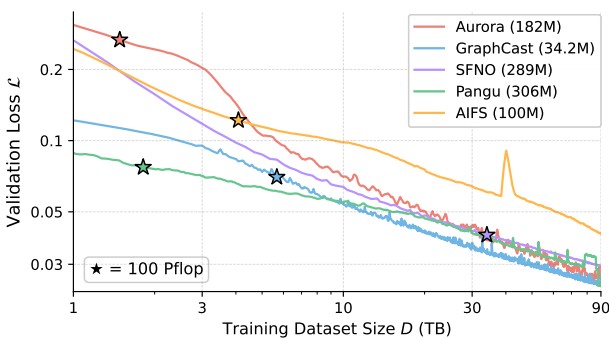

*Figure 1.* **Scaling behavior of global weather models.** We report the validation loss during training to evaluate model performance. We also report the training data required to reach 100 Pflop compute budget.

et al., 2021; Ben Bouallegue et al., 2024). The growing availability of high-resolution observational data and advances in large-scale deep-learning infrastructure have been pushing these models even further. In the near future, it is expected that weather models will undergo a period of rapid growth by scaling toward practical limits.

*Scaling laws* (Kaplan et al., 2020; Hoffmann et al., 2022) have driven great success in large language models (Radford et al., 2019; Brown et al., 2020) and can help improve data-driven weather models. Over the past decade, research has shown that model performance scales predictably with increasing computational resources. Scaling laws describe empirical power-law relationships between validation loss ($\mathcal{L}$) and key training factors: model size (number of parameters, $N$), dataset size ($D$), and total compute budget ($C$). These relationships have proven remarkably consistent across natural language processing (NLP) and computer vision domains (Kaplan et al., 2020):

$$\mathcal{L}(D) = \alpha D^{-\beta}, \quad \mathcal{L}(N) = \gamma N^{-\delta}, \quad \mathcal{L}(C) = \lambda C^{-\epsilon}$$

Here, $D$ represents the cumulative number of training samples ingested during training (i.e., the total volume of training data processed up to that point), not the size of a fixed dataset. This follows the standard convention in scaling law literature (Kaplan et al., 2020; Hoffmann et al., 2022), where $D$ effectively corresponds to training progress measured in samples seen. Researchers use linear regression on a log-log plot to fit the coefficients $\alpha$, $\beta$, $\gamma$, $\delta$, $\lambda$, and $\epsilon$. Such

*Table 1.* Summary of weather models. We report the default or most frequently investigated $N$ in the corresponding papers. Variable counts treat one atmospheric variable with multiple vertical levels as a single variable and only include upper-air and surface variables. Our work presents: $\beta$ is the data-scaling coefficient in $\mathcal{L}(D) = \alpha D^{-\beta}$ and larger is better.

| Model | Release Year | #Parameters | Backbone | Training Loss | #Surface + #Upper-air Variables | $\beta$ ↑ |
|---|---|---|---|---|---|---|
| Aurora | 2024 | 1.3B | Swin Transformer | MAE | $\sim 4 + 5$ | **0.51** |
| AIFS | 2024 | 255M | Graph Transformer | MSE | $\sim 10 + 5$ | 0.46 |
| Pangu | 2023 | 276M | Swin Transformer | MAE | $\sim 4 + 5$ | 0.43 |
| GraphCast | 2023 | 36.7M | Graph Neural Network | MSE | $\sim 5 + 6$ | 0.36 |
| SFNO | 2023 | 433M | Spherical Fourier Operator | MSE | $\sim 8 + 5$ | 0.34 |

laws enable practitioners to predict model performance at scale and optimize resource allocation under a fixed training compute budget $C$, i.e., compute-optimal training.

Several reasons motivate a systematic study of scaling laws of global weather models. On one hand, the growing availability of high-resolution reanalysis data and satellite observations, combined with the rising training cost of large weather models, makes understanding scaling behavior increasingly urgent. Achieving the spatial and temporal resolution of numerical weather prediction systems will demand larger networks and more efficient use of computational resources. On the other hand, applying existing scaling laws in the language and vision domains to weather models faces several fundamental challenges. The atmosphere is a chaotic system with physical limits of predictability (Zhang et al., 2019; Lorenz, 1969). Weather models predict hundreds of correlated yet physically different variables including temperature, wind speed, pressure level, with distinct prediction difficulty. Architectural differences between models also affect scaling behavior, as is already verified in the language and vision domains (Li et al., 2025). Therefore, systematically studying scaling laws in global weather models is essential for efficiently guiding model development.

In this work, we provide the first cross-model scaling analysis in weather prediction (Figure 1), based on establishing the existence and functional form of scaling laws for GraphCast (Lam et al., 2023), AIFS (Lang et al., 2024), Aurora (Bodnar et al., 2024), Pangu (Bi et al., 2023), and SFNO (Bonev et al., 2023) models (see Table 1), covering 3 of 4 models available in the ECMWF ai-models repository (ECMWF Lab, 2024). We benchmark these state-of-the-art models on ERA5 (Hersbach et al., 2020) data under unified experimental conditions, systematically varying $N$, $D$, $C$, and model shape to empirically characterize how forecast skill responds to increased resources. We validate model performance using both validation loss $\mathcal{L}$ and variable-specific loss.

In summary, our contributions and conclusions are:

- We provide the first cross-model analysis of scaling laws that relate forecast skill to $N$ and $D$.

- Aurora achieves better data-scaling efficiency. Graph-Cast demonstrates better parameter-scaling efficiency.

- Scaling behavior in weather forecasting deviates from NLP and vision, with advantage for wide models.

- Scaling behavior varies across variables. The validation loss across all variables is only a rough indicator of overall model performance.

## 2. Methodology

This work aims to reproduce and analyze neural scaling laws in the context of data-driven weather forecasting. We examine how model performance scales with training compute, model size, and dataset volume by pre-training a suite of representative models—GraphCast, AIFS, Aurora, Pangu, and SFNO—on reanalysis data. All models are trained and evaluated under consistent conditions to enable fair cross-model comparisons. Our work builds upon these established models by not only benchmarking their performance but also investigating whether scaling trends are preserved across different models and data regimes.

### 2.1. Model Details

As outlined and summarized in Table 1, this study evaluates a diverse set of models. Within the Swin Transformer family, Aurora and Pangu differ in their feature-extraction pipelines: Aurora applies a patchifying step combined with an initial transformer block, while Pangu uses convolutional layers to encode the input data. AIFS performs attention over nodes in a reduced, more uniformly distributed spherical graph rather than the highly non-uniform latitude–longitude grid, yielding a more balanced global representation and reducing lat–lon distortion. Together, these structural variations create a diverse landscape for evaluating the interplay between models and predictive performance.

Another critical structural distinction between Graph Neural Network (GNN) and their Transformer or Fourier neural-operator counterparts concerns their dependence on spatial resolution. In GNNs, the number of mesh points is independent of input grid resolution. However, mesh resolution still strongly affects memory consumption and communication

during training and inference. Consequently, although the number of GNN mesh points is invariant to grid density, computational throughput remains tightly constrained by the resolution of the input mesh.

Scaling laws study the impact of model shape and parameter count on performance. Model shape comprises two aspects: width and depth. For Transformer-based models, width refers to the model dimension (that is, the embedding dimension or latent-space size), and depth corresponds to the number of transformer blocks. For GNNs, width is likewise defined as the model dimension, and depth as the number of message-passing steps, with each step consisting of one fixed block of layers. For SFNO, width denotes the model dimension and depth the number of Fourier neural-operator blocks. Notably, the model dimension of Transformer equals the product of the number of attention heads and the dimension of each head. Following Aurora (Bodnar et al., 2024) and related studies, our scaling-law experiments vary the number of heads while keeping the head dimension.

The parameter count $N$ is determined by both the model's dimensionality and its specific architecture. In general, increasing the width or depth enlarges the model, but the rate of growth is non-uniform across components. For example, doubling the width causes dense weight matrices to quadruple in size, whereas layer-normalization weights scale linearly. As a result, constructing models with different shapes while keeping $N$ fixed is practically difficult because of architectural constraints, such as the requirement that the width be a multiple of the attention-head dimension. For GraphCast, the parameter count is governed by the width $w$ and depth $d$ according to $N = (24 + 8d)\,w + (18 + 7d)\,w^2$. In this expression, the coefficients are structural constants derived from the network topology and not from data properties. Therefore, to rigorously evaluate the effect of model shape on validation loss, we restrict our analysis to models with comparable $N$.

## 2.2. Training Dataset and Setup

We utilize the ERA5 dataset (Hersbach et al., 2020), provided through WeatherBench 2 (Rasp et al., 2024), which contains atmospheric variables at 6-hour intervals. The data is accessed in the `zarr` format and requires no additional preprocessing. All models are trained to produce 6-hour forecasts from the 00:00, 06:00, 12:00, and 18:00 UTC time points each day.

To ensure comparability across models, we apply the following standardizations: 1) the maximal common set of input and target variables: geopotential, temperature, u/v wind components, specific humidity, etc; 2) consistent spatial resolution ($0.25° \times 0.25°$ global grid); 3) constant learning rate schedules (after optional warm-up stage); 4) identical training dataset (ERA5 1979–2020) and validation dataset

(ERA5 2021). Details regarding the optimizer, learning rate, batch size, parallelism, and model weight initialization are summarized in Appendix B.1. To ensure stable training dynamics across varying scales and models, we conducted hyperparameter sweeps for each model. Given the computational cost of large-scale weather model training, our approach prioritized finding stable learning rates that avoid divergence while maintaining consistent validation loss improvement. In this process, we evaluated the application of Maximal Update Parameterization ($\mu$P) (Yang et al., 2021); however, we observed that $\mu$P is ineffective for training AIFS. Details of the hyperparameter sweeps are provided in Appendix B.2. The total computational cost for these experiments exceeded 430,000 GPU hours.

Finally, to rigorously quantify model performance beyond training objectives, we standardize the evaluation metric used for validation.

## 2.3. Validation Loss Normalization Across Models

We used the same validation loss function in all models to ensure comparability, even though the models employ distinct default training loss functions and variable weightings during both training and validation as shown in Table 1. Our validation loss $\mathcal{L}$ computes the weighted average of squared differences between prediction and ground-truth ($(\hat{x} - x)^2$) across spatial grid cells and atmospheric variables, as defined in Appendix Equation (1). Each variable is normalized by its standard deviation, and each grid cell is weighted by its normalized area to account for the Earth's spherical geometry. For upper-air variables, an additional weight, proportional to the pressure level, is applied—similar to Graphcast—resulting in higher pressure levels being more important (Lam et al., 2023). Details regarding variable units and weights are provided in Appendix D.

Overall, these strict standardization protocols minimize the influence of extrinsic factors such as data distribution shifts or optimization inconsistencies. By aligning the training configurations, input modalities, and initialization baselines, we ensure that any divergent performance metrics observed in the subsequent analysis can be attributed to the intrinsic architectural characteristics of the respective models. This unified framework provides a robust foundation for evaluating the efficacy of distinct design choices—from the spectral handling in SFNO to the mixed initialization schemes of Aurora—under identical experimental conditions.

## 3. Results

This section presents our analysis of scaling laws in data-driven weather forecasting. We investigate fundamental scaling relationships across $N$ and $D$. In contrast to existing scaling laws (Kaplan et al., 2020; Hoffmann et al., 2022;

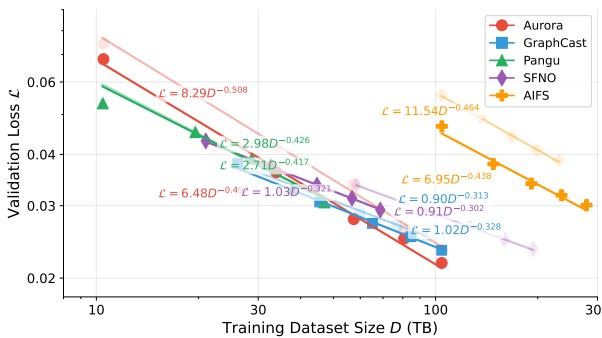

*Figure 2.* **Data-scaling laws** across weather forecasting models: $\mathcal{L}(D) = \alpha D^{-\beta}$. Aurora (red) achieves the best $\mathcal{L}$ at $D = 100$ TB and also has the best $\beta$ value, representing most efficient scaling with more data. The darker markers represent the larger models. Model specifications are provided in Appendix Table 6.

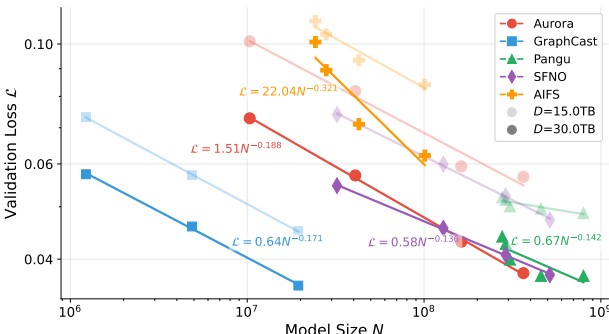

*Figure 3.* **Parameter-scaling laws** for different weather prediction models trained on 15.0 TB and 30.0 TB datasets. Each marker corresponds to a model variant, with dashed lines denoting power-law fits of the form $\mathcal{L}(N) = \gamma N^{-\delta}$. Marker transparency indicates dataset size, with darker markers representing larger $D$. At fixed $D$, validation loss improves with larger $N$, and increasing $D$ shifts the scaling curves downward while preserving or increasing the scaling exponent $\delta$ within each model.

Zhai et al., 2022), model shape plays a significant role in weather models. Then we explain why compute-optimal scaling of weather models is different from language models. We also compare scaling performance of different variables.

### 3.1. Power Law of Scaling with $D$ and $N$

We observe that validation loss follows power-law relationships with $D$ and $N$ (Kaplan et al., 2020). We first examine data-scaling laws of the form $\mathcal{L}(D) = \alpha D^{-\beta}$. For each model configuration with parameter count $N$, we train using different values of $D$ and fit the scaling factors $\alpha$ and $\beta$. A smaller $\alpha$ indicates lower validation loss at the early stage, while a larger $\beta$ implies that validation loss decreases more rapidly and is therefore more advantageous asymptotically.

As illustrated in Figure 2, the data-scaling laws vary across models and specific values of $N$. AIFS ($\beta \approx 0.46$) and Pangu ($\beta \approx 0.43$) show higher sensitivity, while Aurora demonstrates the steepest scaling with $\beta \approx 0.51$. These variations reflect the differing efficiencies with which each model extracts information from training data. Conversely, comparing the intercept $\alpha$ directly is challenging, as the ranking of models by validation loss depends on the given $D$. For instance, while GraphCast achieves the lowest validation loss at $D = 30$ TB, Aurora outperforms it at $D = 100$ TB. Ultimately, $\beta$ serves as a better indicator of long-term potential. The higher $\beta$ observed in Aurora ($\beta \approx 0.51$) compared to other models ($\beta$ ranging from 0.30 to 0.46) indicates that **Aurora is the most efficient model for scaling with increasing dataset sizes**.

Next, we study model-scaling laws of the form $\mathcal{L}(N) = \gamma N^{-\delta}$. We fix the training dataset size and fit the validation losses of models with different $N$. A smaller $\gamma$ indicates that only a modest number of parameters is required to achieve low initial loss, whereas a larger $\delta$ implies that performance improves quickly as $N$ increases and thus becomes more

advantageous when $N$ is scaled to large values. Collectively, these findings highlight systematic differences in how models exploit additional data or parameters, providing a basis for cross-model comparison.

As shown in Figure 3, all five models exhibit consistent power-law scaling behavior across both 15.0 TB and 30.0 TB dataset sizes, confirming that the relationship $\mathcal{L}(N) = \gamma N^{-\delta}$ robustly characterizes parameter-performance trade-offs in weather prediction models. **GraphCast achieves lower validation loss at equivalent or smaller $N$**, demonstrating superior parameter efficiency; however, it incurs similar computational cost per training step due to its message-passing mechanism. Doubling the dataset size shifts the scaling curves downward while preserving the exponents, suggesting that increasing $D$ and $N$ provide complementary benefits.

The results for Pangu reveal a threshold effect where dataset size determines whether parameter-scaling is effective. At 15.0 TB, Pangu's parameter-scaling exponent $\delta$ is lower than other models, indicating limited benefit from additional parameters. At 30.0 TB, $\delta$ increases, demonstrating that Pangu's capacity to leverage additional parameters depends on sufficient training data. This demonstrates that parameter-scaling is constrained by data volume, and that scaling behaviors change once sufficient data is available.

In summary, our complete analysis reveals that while specific exponents may vary locally, the relationship between $\mathcal{L}$, $N$, and $D$ holds firm. The improved efficiency of Pangu's parameter-scaling ($\delta$) in data-rich regimes serves as evidence of this coupling. Rather than indicating divergent behaviors, these results confirm that **scaling laws for $D$ and $N$ remain consistent and predictable across diverse models and computational scales.**

## 3.2. New Findings on Scaling Laws: Weather Models Favor Wider Configurations

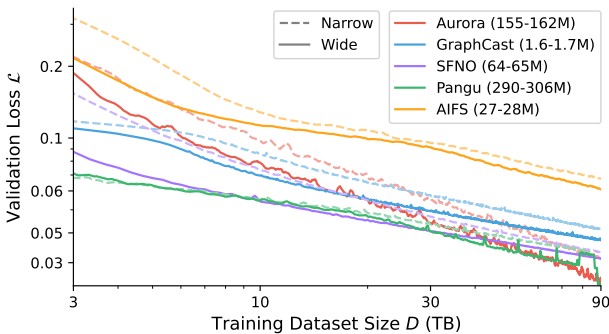

*Figure 4.* **Wider models perform better.** For each model, we use two configurations: one is wider and one is narrower. They have roughly the same $N$ but different shapes. In all models, wider configurations consistently achieve lower validation loss. This implies weather forecasting benefits more from representational capacity (width) than from additional nonlinear transformations (depth).

In addition to studying the overall impact of $N$ on performance, we conduct targeted experiments to evaluate the role of model shape. Unlike Transformer-based language models, where performance depends very mildly on model shape when $N$ is held fixed (Kaplan et al., 2020), we find that increasing width is consistently more efficient than increasing depth—a trend that holds across multiple models. To systematically evaluate the width–depth trade-off, we identify model configurations with similar parameter counts across models and compare their performance under different width–depth ratios.

As shown in Figure 4, we consistently observe across all models that wider configurations outperform narrower alternatives when parameter counts are closely matched. For Aurora, the wide configuration achieves superior performance compared to the narrow variant, despite similar parameter budgets. In GraphCast, increasing width (width=256) with fewer message-passing steps consistently outperforms narrower configurations (width=128) with more steps at the 1.6M-parameter scale. Pangu's wider variant likewise demonstrates better scaling characteristics than its deeper counterpart. SFNO exhibits similar behavior, where the wide variant is consistently superior to the narrow one.

Regarding the optimal aspect ratio, our results indicate that shallower networks are effective. Notably, both GraphCast and SFNO perform well even at a depth of 1. This observation aligns with established findings in geometric deep learning, where simplified, linear graph architectures have been shown to match the performance of deeper counterparts, demonstrating that the essential topological aggregation can be achieved without deep non-linear stacking (Wu et al., 2019; Li et al., 2018). Furthermore, in the context of neural

operators, theoretical works suggest that single-layer spectral or attention mechanisms can effectively approximate complex global optimization steps, rendering deep stacks redundant for certain physical dynamics (Von Oswald et al., 2023). Collectively, these findings imply that the default depths employed in these models may be greater than necessary, and suggest that the short-term dynamics of 6-hour weather prediction are approximated by linear models.

Taken together, these experiments suggest that **width should be prioritized over depth** when designing weather forecasting models under a fixed parameter budget. This conclusion stands in contrast to prior NLP scaling law results (Kaplan et al., 2020). The optimal width–depth ratios we observe highlight a key difference: weather forecasting benefits more from representational capacity (width) than from additional nonlinear transformations (depth).

### 3.3. Compute-Optimal Training

From scaling law studies in the NLP domain (Kaplan et al., 2020; Hoffmann et al., 2022), we know that $C$, $N$, and $D$ should scale in tandem to perform the most efficient training. The trade-off between using larger $N$ and larger $D$ given a fixed $C$ has always been the main concern of scaling laws. For a fixed $C$, there exists an optimal allocation between parameters and training steps. Training a model that is too large (under-trained) or too small (capacity-limited) both results in suboptimal performance. When fitting validation loss against $C$ using parabolas (Hoffmann et al., 2022), we observe a trade-off between $N$ and $D$. This trade-off can be formalized as $N_{\text{opt}} \propto C^a$ and $D_{\text{opt}} \propto C^b$, where $N_{\text{opt}}$ is the optimal number of parameters, $D_{\text{opt}}$ is the optimal number of training steps, and $C$ is the total compute budget.

In contrast to Transformer-based language models (Kaplan et al., 2020), the scaling constraint $a + b = 1$ does not strictly hold for all weather forecasting models. For models like GraphCast and AIFS, the backbone operates on a latent graph that is independent of the input spatial resolution, violating the standard assumption that $C \approx 6ND$. However, because the resolution in our experiments is fixed, the compute cost scales linearly with parameter count and batch size: $C \sim N \cdot B$. Conversely, in Transformer-based models, the patching mechanism reduces the effective $D$ by a factor of $p^2$. This yields the modified equation $C \approx 6ND/p^2$, where the constraint $a + b = 1$ is maintained.

As shown in Figure 5, the compute-optimal scaling behavior of architectural families under a fixed flop budget vary. The locations of the minima yield the compute-optimal splits between parameters and data. Practically, as compute grows, the optimal allocation still favors increasing training dataset over $N$. For Aurora and Graphcast, we observe that the parabolas appear primarily in the left half. This is because the models have not yet reached convergence and smaller

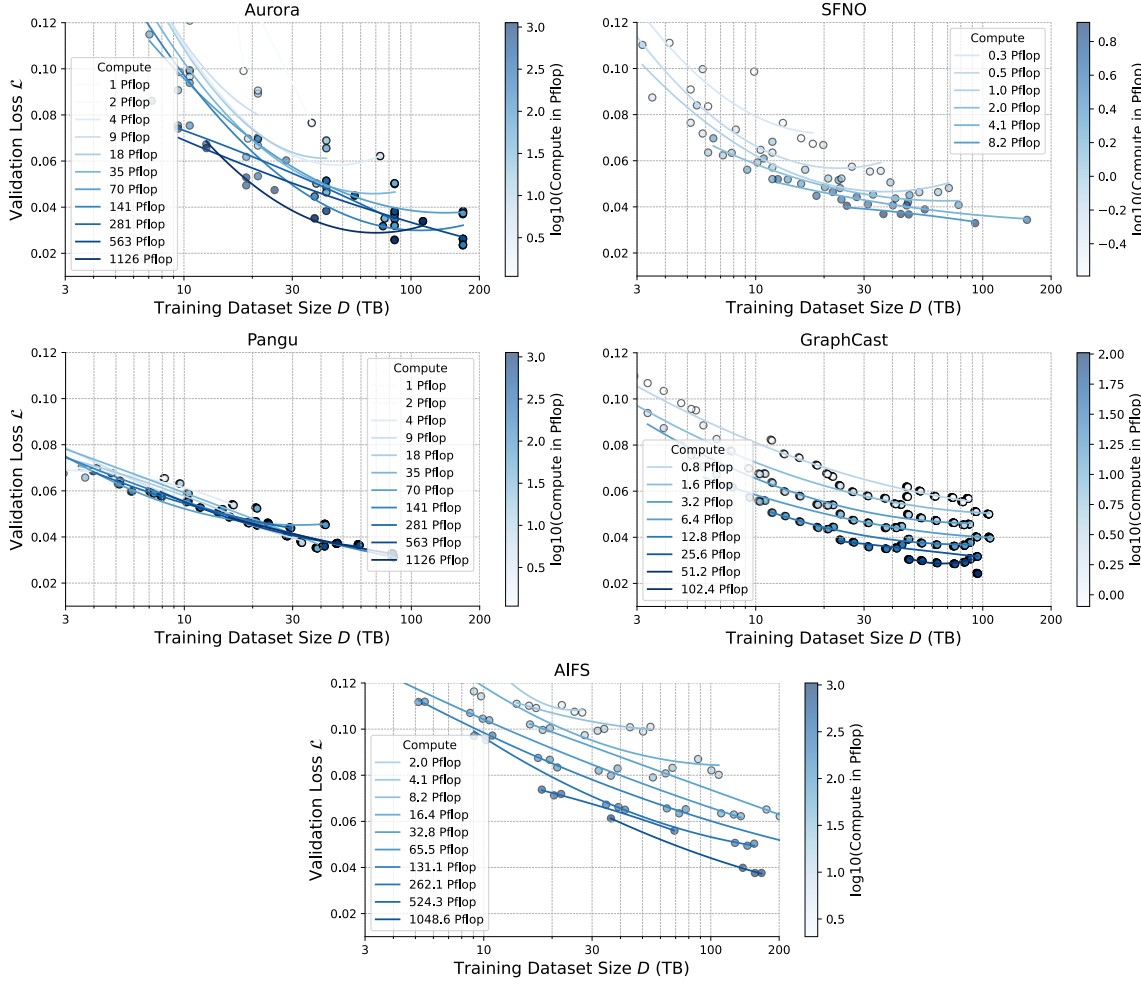

*Figure 5.* **Compute-optimal training.** The panels illustrate $\mathcal{L}$ as a function of $D$, with each curve representing a fixed $C$. The resulting parabolas identify the compute-optimal frontier—the specific ratio of $N$ and $D$ that minimizes loss for a given $C$. For each model, we have $C \sim ND$, which implies $N \sim C/D$ along each curve. For Pangu, the loss is predominantly determined by $D$. Aurora and SFNO exhibit a clear minimum in the curve, indicating a trade-off between larger $N$ and larger dataset size. GraphCast and AIFS shows mainly the left half of the parabolas, indicating performance is bottlenecked by insufficient $D$.

models need significantly more training time when flop per step is less than 10% of larger ones. Also, the minimum width for Aurora is 64 (the size of one attention head), and from current tendency, the right half of the parabolas need even smaller width. However, for Pangu, we do not observe a clear relationship between $\mathcal{L}$ and $C$. In contrast, validation loss shows clear power-law dependency on training data. For SFNO, the pattern is similar to Aurora, with the parabolas mostly in the left half.

Memory constraints also limit our exploration of the full compute-optimal frontiers, with a maximum achievable width of 512. Despite these limitations, the consistent parabola behavior across models confirms the universality of compute-optimal trade-offs in weather forecasting models, albeit with model-specific optimal ratios. This analysis

indicates that for practical application, **as compute budgets increase, the optimal allocation strategy still favors increasing $D$ over $N$.**

### 3.4. Variable-Specific Scaling Performance

Our analysis reveals heterogeneity in how different atmospheric variables benefit from increased model capacity. Not all meteorological variables scale equally within the same model. We observe clear differences in how effectively different atmospheric variables benefit from increased model capacity, with some variables showing steeper or less smooth scaling curves than others. To illustrate this, we focus on the RMSE of 10m u-component of wind (10U) and 2m temperature (2T), validated on 2021 data, comparing all five models across varying widths and depths.

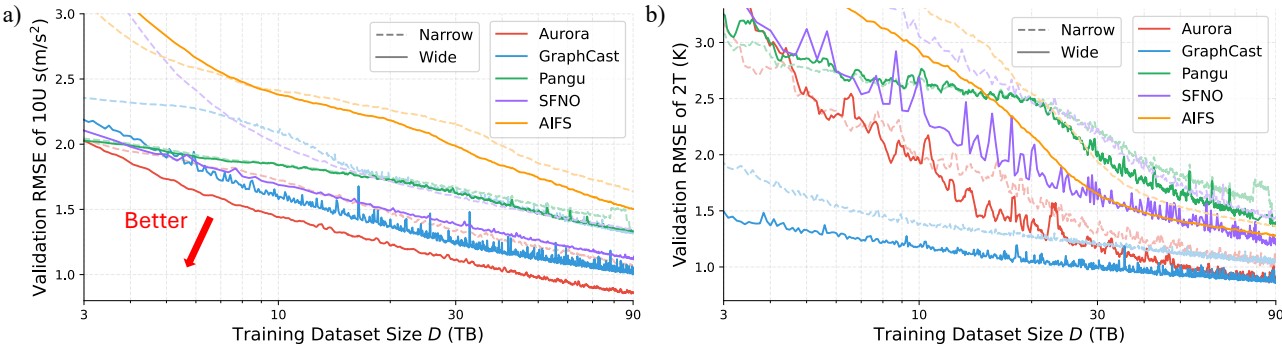

*Figure 6.* **Variable-specific loss scaling for 10m u-component of wind (10U) and 2m temperature (2T)**. We compare Aurora, GraphCast, Pangu, SFNO, and AIFS. Validation RMSE is plotted against the volume of training data in TB. Each line represents a different model configuration, with line color indicating $N$ (lighter colors for narrower models, darker for wider). For 10U, Aurora achieves the lowest RMSE at all dataset sizes. For 2T, GraphCast's performance is highly competitive and surpasses all models.

For the 10U variable in Figure 6(a), Aurora achieves the lowest RMSE for larger widths, outperforming both Graph-Cast and Pangu. Pangu improves with both width and depth scaling, but does not reach Aurora's best results. For the 2T variable in Figure 6(b), a similar trend emerges, though GraphCast's performance surpasses that of Aurora. Graph-Cast is also more competitive at smaller $N$ and converges faster in early training.

In conclusion, **variables do not scale equally**, exhibiting heterogeneity in their response to increased $N$ and $D$. Aurora achieves the lowest RMSE for 10U and lowest weighted loss, whereas GraphCast's performance for 2T is the best. Therefore, the weighted loss across all variables is only a rough indicator of overall model performance and researchers must take this variable-specific scaling into account.

## 4. Discussion

Our experiments demonstrate that across all tested models, increasing width consistently yields better performance than increasing depth when parameter counts are fixed. Both GraphCast and SFNO perform well even at depth=1, suggesting that 6-hour weather prediction dynamics are well-approximated by linear models (Wu et al., 2019; Li et al., 2018) and that single-layer architectures can effectively approximate the relevant predictive operators (Von Oswald et al., 2023). This finding suggests that weather forecasting relies more on high representational capacity than deep sequential reasoning.

Scaling laws appear to reveal distinct efficiency profiles reflecting differences between models. GraphCast relies on GNN-based message passing with Multi-Layer Perceptrons, avoiding the quadratic costs of attention but potentially introducing communication overhead that may constrain practical scaling. Aurora's relatively superior scaling

efficiency may stem from unified 3D tokenization preserving vertical atmospheric coupling (Bodnar et al., 2024), whereas Pangu's variable splitting may introduce additional complexity. Aurora's attention-based embedding provides immediate global receptive fields, unlike Pangu's convolutional downsampling restricted to local neighborhoods. SFNO leverages Fourier operators for frequency-domain processing, achieving robust performance even at depth=1. These mechanical differences appear to influence both current performance and scaling trajectories.

Computational efficiency reveals critical differences in hardware utilization. As shown in Table 2[1], while GraphCast achieves the best validation loss under the same $N$, it utilizes only 1.03% of the single precision peak on H100 GPUs (NVIDIA Corporation, 2022). In contrast, Aurora achieves 37.2% utilization, which is approximately $36\times$ of GraphCast. This performance gap stems from model differences: GraphCast's GNN relies on message-passing operations, which are memory-bounded and less optimized than the transformer attention mechanisms used by Aurora (Deng & Rao, 2024). These findings highlight a limitation in classical scaling law analyses (Kaplan et al., 2020; Hoffmann et al., 2022), which typically treat compute as a static resource and ignore wall-clock time. Furthermore, engineering optimizations such as hardware-specific implementations, memory management, and parallelization strategies are explicitly excluded from scaling laws, yet these factors critically determine practical model deployment efficiency.

For resource allocation, our compute-optimal analysis indicates operational systems should prioritize smaller models trained for longer durations over big models with insufficient training. In other words, for a fixed compute budget, allocating resources toward extending training steps yields

---

[1]The limited performance of SFNO and Pangu is mainly caused by dataloading bottleneck. With synthetic data, their peak performance reaches 9.56 Tflop/s and 33.5 Tflop/s respectively.

*Table 2.* **Computational efficiency of different models on H100 GPUs.** Peak performance during training varies by precision: AIFS uses 16-bit half precision, while others use 32-bit single precision. Utilization is calculated relative to each model's corresponding peak performance.

| Model | Tflop/s | GPU peak (Tflop/s) | GPU util. (%) |
|---|---|---|---|
| Aurora | 368 | 989 (32-bit) | 37.2 |
| AIFS | 33.7 | 1979 (16-bit) | 1.70 |
| GraphCast | 10.15 | 989 (32-bit) | 1.03 |
| Pangu | 3.25 | 989 (32-bit) | 0.33 |
| SFNO | 0.215 | 989 (32-bit) | 0.022 |

a greater reduction in forecast error than allocating them toward additional parameters whose benefits cannot be realized without sufficient optimization. This finding aligns with observations in language models (Hoffmann et al., 2022).

Beyond aggregate validation loss, variable-specific analysis reveals heterogeneous scaling responses. Performance does not improve uniformly with increased $N$ and $D$, and the relative ordering of models varies across variables. These disparities stem both from model design biases—such as how different models represent spatial locality, anisotropy, and multi-scale interactions—and from the intrinsic difficulty of the variables themselves. These heterogeneous behaviors suggest variable-specific tuning or hybrid modeling strategies to fully capitalize on scaling improvements.

Together, these findings reveal that weather forecasting models operate under different scaling principles than NLP, with width preferred over depth, distinct model efficiency profiles, and critical hardware utilization constraints that limit practical deployment. The heterogeneous variable-specific scaling behaviors further demonstrate that optimal model design must balance theoretical efficiency, practical hardware constraints, and domain-specific physical requirements.

## 5. Related Works

Scaling laws have been established across multiple domains, demonstrating predictable power-law relationships between model performance and key factors such as model size ($N$), dataset size ($D$), and compute budget ($C$). Kaplan et al. (2020) established the foundational framework for neural scaling laws in language modeling, deriving compute-optimal allocation rules where $N \propto C^{0.73}$ and $D \propto C^{0.27}$ for fixed compute budgets. Hoffmann et al. (2022) later refined this view by proposing compute-optimal scaling with adjusted exponents. Similar scaling relationships have been observed in vision (Zhai et al., 2022), reinforcement learning (Neumann & Gros, 2022), and other domains (Li et al., 2025; Villalobos, 2023; Hernandez et al., 2021), confirming that scaling behavior is robust and predictable across diverse machine learning applications.

However, systematic scaling law analyses for weather forecasting models remain limited, even though they can efficiently guide model development. Key models include GraphCast (Lam et al., 2023), which employs graph neural networks; AIFS (Lang et al., 2024), utilizing graph transformers; Pangu-Weather (Bi et al., 2023), based on Swin Transformer; FourCastNet (Pathak et al., 2022) and SFNO (Bonev et al., 2023), which leverage Fourier-based operators; and Aurora (Bodnar et al., 2024), a foundation model utilizing hierarchical attention. Subsequent research has expanded these approaches into kilometer-scale resolution (Han et al., 2024) and probabilistic diffusion modeling (Price et al., 2023). All of these models rely on reanalysis datasets such as ERA5 (Hersbach et al., 2020). While there are some works exploring scaling laws for individual weather models (Nguyen et al., 2024; Couairon et al., 2026), most studies focus on single model configurations rather than characterizing scaling behavior across models.

## 6. Conclusion

In conclusion, our findings establish a quantitative framework for atmospheric prediction and reveal scaling behaviors that differ from those in language models. Across all five models, we find that model width is more decisive than depth; wide configurations consistently achieve lower validation loss at fixed parameter counts, offering superior performance and practical advantages in training speed. This stands in sharp contrast to Transformer-based language models, where performance depends very mildly on model shape at fixed $N$.

The scaling analysis reveals distinct model strengths: Aurora achieves the strongest scaling with dataset size $D$, with $\beta \approx 0.51$ compared to 0.30–0.46 for other models. Parameter-scaling ($N$) reveals a nuanced distinction between theoretical and practical efficiency. While GraphCast theoretically achieves superior parameter efficiency, its practical scalability is constrained by the memory-bandwidth bottlenecks inherent to message-passing operations. Our compute-optimal analysis indicates that as compute budgets increase, the optimal allocation strategy consistently favors increasing training duration over model size. Additionally, we identify heterogeneity in variable-specific scaling behavior, implying that overall weighted loss provides only a rough indicator of model performance. Finally, our results demonstrate that superior validation loss scaling does not guarantee efficient compute utilization. This is starkly illustrated by Aurora achieving approximately $36\times$ higher hardware utilization than GraphCast, underscoring the impact of engineering optimizations on practical deployment.

We suggest that future weather models should prioritize wider architectures and larger training datasets to maximize predictive performance.

## Software and Data

The source code for our experiments is available at https://github.com/spcl/scaling-laws-weather-model. The dataset we use is ERA5 (Hersbach et al., 2020) provided through WeatherBench 2 (Rasp et al., 2024).

## Acknowledgements

This work was supported by the WeatherGenerator project (grant agreement No. 101187947), the ERC PSAP project (grant agreement No. 101002047), and under project ID a01 and a122 as part of the Swiss AI Initiative, through a grant from the ETH Domain and computational resources provided by the Swiss National Supercomputing Centre (CSCS) under the Alps infrastructure.

The authors would like to thank Yun Cheng, Firat Ozdemir, and Salman Mohebi for their work on the Aurora trainer implementation, and Joel Oskarsson for his constructive feedback and comments on the manuscript.

## Impact Statement

This paper presents work whose goal is to advance the field of Machine Learning, particularly its applications to data-driven weather modeling. Our study aims to improve understanding of how scaling behavior influences model performance and computational efficiency in large-scale Earth system prediction. By bridging advances in machine learning with atmospheric science, we hope to contribute to the development of more accurate, efficient, and accessible weather forecasting systems. There are many potential societal consequences of our work, including improved preparedness for extreme events and better resource management.

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

# A. Training Implementation

All models in this work are implemented with PyTorch or JAX backends. To ensure reproducibility, we containerize the environment setup with Dockerfiles specifying CUDA, NCCL, and model-specific dependencies. For GraphCast, we adopt the official JAX implementation from Google DeepMind[2], but find that significant debugging and reimplementation of the training pipeline are required. In particular, we replace the Colab-based demonstration with a full Python script, add explicit gradient all-reduce, checkpointing with optimizer state, and support for multi-node training. Aurora is based on the official Microsoft implementation[3], but we use a re-implementation[4] of the training loop in PyTorch Lightning and a new stateful dataloader to enable flexible checkpointing and reliable resumption (Ozdemir et al., 2026). Pangu relies on a PyTorch reimplementation[5], with its earlier pseudo-code version (Bi et al., 2023)[6] consulted but not sufficient for reproducibility. To ensure experimental consistency, we use the same stateful dataloader as Aurora that supports mid-epoch resumption, alongside modifications for distributed training. SFNO is reproduced using NVIDIA's Makani repository[7]; although the base implementation is stable, we re-implement its dataloader in PyTorch to ensure statefulness and consistency with the other models. AIFS is provided in Anemoi package[8] from ECMWF.

We developed a unified data pipeline across all models using ERA5 reanalysis data in Zarr format. Following Weather-Bench (Rasp et al., 2024) conventions, we use a 6-hour timestep and restrict the training set to the 0, 6, 12, and 18 UTC hours of each day. This choice balances temporal coverage with computational feasibility while ensuring comparability across models.

Checkpointing is implemented in all models to save both parameters and optimizer states at regular intervals. All PyTorch-based models (Aurora, AIFS, Pangu, SFNO) require custom stateful dataloaders or saving the index of used data batches to support reliable checkpointing, while GraphCast's JAX pipeline demands a full reimplementation of training to achieve consistent multi-node behavior.

To explore scaling behavior, models are modified in width and depth, while holding other factors constant. For GraphCast, we vary latent size and message-passing steps. For Aurora, we systematically adjust width and depth for every layer in the encoder and decoder of the Swin Transformer backbone. For Pangu, we modify width and depth within the constraints of the official implementation, which splits surface and upper-air variables. For SFNO, we change the embedding dimension and number of operator layers. For AIFS, we vary the latent size as width, and the number of transformer blocks in the Transformer backbone as depth.

# B. Training Configuration and Hyperparameter Tuning

To ensure transparency and reproducibility, we provide full details regarding our training configurations and hyperparameter sweeps. All models were trained using Distributed Data Parallel (DDP) with either Adam or AdamW optimizers.

### B.1. Training Configuration Summary

Table 3 summarizes the final hyperparameter configurations used for our experiments.

Regarding model weights, we utilize default initialization strategies, which vary by models: SFNO uses a custom scaled normal approach, while AIFS and GraphCast rely on their respective framework defaults (PyTorch and JAX/Haiku). Pangu applies a uniform truncated normal initialization, and Aurora utilizes a mixed strategy of Kaiming (He et al., 2015) and Xavier (Glorot & Bengio, 2010) uniform distributions, depending on the layer type.

### B.2. Hyperparameter Sweep Details

We conducted targeted hyperparameter sweeps for each weather model as follows:

---

[2]https://github.com/google-deepmind/graphcast
[3]https://github.com/microsoft/aurora
[4]https://github.com/swiss-ai/ESFM
[5]https://github.com/zhaoshan2/pangu-pytorch
[6]https://github.com/198808xc/Pangu-Weather
[7]https://github.com/NVIDIA/makani
[8]https://github.com/ecmwf/anemoi-core

*Table 3.* **Training configurations for all models.**

| Model | Optimizer | Learning rate | Betas | Parallelization | Batch size |
|---|---|---|---|---|---|
| Aurora | AdamW | $5 \times 10^{-4}$ | (0.9, 0.999) | DDP | 16 |
| GraphCast | Adam | $1 \times 10^{-4}$ | (0.9, 0.999) | DDP | 16 |
| Pangu | Adam | $5 \times 10^{-4}$ | (0.9, 0.999) | DDP | 4 |
| SFNO | AdamW | $1 \times 10^{-4}$ | (0.9, 0.95) | DDP | 16 |
| AIFS | AdamW | $1 \times 10^{-4}$ | (0.9, 0.95) | DDP | 16 |

- **Aurora:** We swept learning rates of $3 \times 10^{-6}, 1.2 \times 10^{-5}, 3 \times 10^{-5}, 1 \times 10^{-4}$, and $5 \times 10^{-4}$, as well as batch sizes of 16 and 32 (at $5 \times 10^{-4}$), evaluated over one 6-hour rollout step for more than 3000 steps. By following the original paper for settings such as linear warmup and patch size of 4 for Swin Transformer, $5 \times 10^{-4}$ provided a balance between the rate of loss reduction and training stability.

- **AIFS:** Following the original scaling rules, we used a local learning rate of $6.25 \times 10^{-6}$, yielding an effective learning rate of $1 \times 10^{-4}$ for a batch size of 16. We additionally evaluated Maximal Update Parametrization ($\mu$P) (Yang et al., 2021) on the largest model (width 1024) with an adjusted rate of $0.707 \times 10^{-4}$ over more than 3000 training steps (corresponding to over 40 TB of dataset size), but observed consistently degraded validation loss curves. Furthermore, the $1 \times 10^{-4}$ effective rate caused loss spikes on the largest model, confirming we were operating near the maximum stable boundary.

- **GraphCast:** While the original paper utilized $1 \times 10^{-3}$, we swept $1 \times 10^{-4}, 3 \times 10^{-4}$, and $1 \times 10^{-3}$ for our batch size of 16. Rates of $1 \times 10^{-3}$ and $3 \times 10^{-4}$ were unstable and diverged early (between 1000–2000 steps); consequently, we scaled down to $1 \times 10^{-4}$, which stabilized the loss curve.

- **Pangu:** We tested the learning rate of $5 \times 10^{-4}$ specified from the original paper for a batch size of 4. Under this configuration, the validation loss exhibited a steady downward trend throughout the training process, rendering further hyperparameter ablations unnecessary.

- **SFNO:** We swept a range of learning rates of $1 \times 10^{-5}, 1 \times 10^{-4}$, and the original paper's $1 \times 10^{-3}$ for a batch size of 16. The original $1 \times 10^{-3}$ proved unstable for our setup, while $1 \times 10^{-4}$ provided the best downward trend without instability.

Finally, we note that exhaustive hyperparameter grid searches are computationally prohibitive at this scale. Moreover, we observed that training these weather models is sensitive to the learning rate. Excessive rates resulted in validation loss divergence even when training loss remained stable. Furthermore, tuning heuristics derived for Large Language Models (such as $\mu$P) do not directly transfer, as fitting continuous weather data differs from predicting discrete text tokens. Compounding this, each weather model exhibits different hyperparameter preferences, requiring configurations tailored to their original papers. Nevertheless, our empirical tuning achieved stable training dynamics with a consistently decreasing validation loss across the evaluated models.

## C. Compute Accounting

Training compute is calculated as:

$$C(\text{model}) = \text{FLOPs/step}_{\text{train}}(\text{model}) \times \#\text{steps},$$

For GraphCast, at multi-mesh level $\ell$, let the grid have $N_g$ nodes, the mesh $N_m(\ell)$ nodes, and the multilevel structure $E_m(\ell)$ edges. Let the model width be $W$, and let there be $S$ message passing steps, which is the depth in model shape.

Grid→Mesh projection applies linear maps to grid nodes, mesh nodes, and edges. Each linear map has terms linear and quadratic in $W$, and multiplies with the number of elements. With multiply-add counted as 2 FLOPs:

$$\text{FLOPs}_{\text{G2M}}^{\text{fwd}} = 2\Big[(4W + 2W^2)N_g + (4W + 3W^2)N_m(\ell) + (4W + 4W^2)E_m(\ell)\Big].$$

The mesh GNN repeats $S$ message passing steps , each updating nodes and edges with parameters scaling linearly and quadratically in $W$:

$$\text{FLOPs}_{\text{Mesh}}^{\text{fwd}} = 2S\Big[(4W + 3W^2)N_m(\ell) + (4W + 4W^2)E_m(\ell)\Big].$$

Mesh→Grid projection reverses the process:

$$\text{FLOPs}_{\text{M2G}}^{\text{fwd}} = 2\Big[(4W + 2W^2)N_m(\ell) + (4W + 3W^2)N_g + (12W + 4W^2)E_m(\ell)\Big].$$

The total forward FLOPs are the sum of these three terms; training FLOPs are three times this.

For Aurora, let the spatial grid be $H \times L$, patch size $p$, and effective channel count $C$. The token count before windowing is

$$n = C \cdot (H/p) \cdot (L/p).$$

Each Swin block has window size $w$, number of windows $n_w = \lceil n/w \rceil$, model width $W$, and $h$ heads with per-head width $W/h$.

Self-attention consists of QKV projection, query–key multiplication, softmax, attention–value multiplication, and output projection:

$$\text{FLOPs}_{\text{attn}}^{\text{fwd}} = 2nW(3W) + 2n_w h\left(w \tfrac{W}{h} w\right) + 5n_w hw^2 + 2n_w h\left(w^2 \tfrac{W}{h}\right) + 2nW^2.$$

The MLP expands hidden size by a factor $r = 4$:

$$\text{FLOPs}_{\text{mlp}}^{\text{fwd}} = 2nW(rW) + 6n(rW) + 2n(rW)W.$$

Two LayerNorms per block contribute

$$\text{FLOPs}_{\text{norm}}^{\text{fwd}} = 10nW.$$

Stage transitions and patch embedding/recovery use

$$\text{FLOPs}_{\text{proj}}^{\text{fwd}} = 2n_{\text{stage}}W_{\text{in}}W_{\text{out}} + 5n_{\text{stage}}W_{\text{in}}.$$

Summing across encoder, backbone, and decoder depths yields forward FLOPs; training FLOPs are three times forward.

For Pangu, a four-stage Swin Transformer, stage $s$ has $n_s$ tokens, model width $W_s$, and $h_s$ heads with head width $W_s/h_s$. The spatial grid at input is $H \times L$, with downsampling and upsampling stages reducing and restoring token counts. With window size $w$, one block in stage $s$ has attention

$$\text{FLOPs}_{\text{attn},s}^{\text{fwd}} = 2n_s W_s(3W_s) + 2\lceil n_s/w \rceil h_s\left(w \tfrac{W_s}{h_s} w\right) + 5\lceil n_s/w \rceil h_s w^2 + 2\lceil n_s/w \rceil h_s\left(w^2 \tfrac{W_s}{h_s}\right) + 2n_s W_s^2,$$

MLP

$$\text{FLOPs}_{\text{mlp},s}^{\text{fwd}} = 2n_s W_s(4W_s) + 6n_s(4W_s) + 2n_s(4W_s)W_s,$$

and normalization

$$\text{FLOPs}_{\text{norm},s}^{\text{fwd}} = 10n_s W_s.$$

Transitions between stages (down or up) are linear projections:

$$\text{FLOPs}_{\text{proj}}^{\text{fwd}} = 2n_{\text{in}}W_{\text{in}}W_{\text{out}} + 5n_{\text{in}}W_{\text{in}}.$$

Patch embedding and recovery follow the same form. Summing across the four stages and two transitions gives forward FLOPs; training FLOPs are three times forward.

For SFNO, the encoder maps $C$ channels to width $W$ with Conv–GELU–Conv:

$$\text{FLOPs}_{\text{enc}}^{\text{fwd}} = 2CW H_{\text{hi}}W_{\text{hi}} + 6W H_{\text{hi}}W_{\text{hi}} + 2W^2 H_{\text{hi}}W_{\text{hi}}.$$

A Fourier block at low resolution applies normalization, spherical transforms, spectral multiplication, and an MLP. With constant $\alpha$ for transform complexity:

$$\text{FLOPs}_{\text{block}}^{\text{fwd}} = 5WH_{\text{lo}}W_{\text{lo}} + \alpha WH_{\text{lo}}W_{\text{lo}} \log_2 \max(H_{\text{lo}}, W_{\text{lo}})$$
$$+ W^2 \ell_{\max} m_{\max} + \alpha WH_{\text{lo}}W_{\text{lo}} \log_2 \max(H_{\text{lo}}, W_{\text{lo}})$$
$$+ \big(2W(2W)H_{\text{lo}}W_{\text{lo}} + 6(2W)H_{\text{lo}}W_{\text{lo}} + 2(2W)WH_{\text{lo}}W_{\text{lo}}\big).$$

The decoder maps back to $C$ channels:

$$\text{FLOPs}_{\text{dec}}^{\text{fwd}} = 2W^2 H_{\text{hi}}W_{\text{hi}} + 6WH_{\text{hi}}W_{\text{hi}} + 2WCH_{\text{hi}}W_{\text{hi}}.$$

If a skip is used, add $2C^2 H_{\text{hi}}W_{\text{hi}}$. Summing encoder, blocks, and decoder gives forward FLOPs; training FLOPs are three times forward.

For AIFS, let the data grid have $N_g$ nodes and the hidden (reduced) graph have $N_h$ nodes. Let $E_{\text{enc}}$ denote encoder edges (grid$\rightarrow$hidden) and $E_{\text{dec}}$ decoder edges (hidden$\rightarrow$grid). Let $W$ be the model width, $D$ the processor depth (number of transformer layers), and $r = 4$ the MLP expansion ratio.

The encoder projects grid nodes to hidden nodes via graph attention. Node embeddings transform input channels $C$ to width $W$, and edge features of dimension $d_e$ are projected similarly:

$$\text{FLOPs}_{\text{enc}}^{\text{fwd}} = 2\Big[ CWN_g + 12WN_h + d_e WE_{\text{enc}} + 4W^2 N_g + 3W^2 N_h + WE_{\text{enc}} + 2rW^2 N_h \Big].$$

The processor applies $D$ transformer layers on the hidden graph. Each layer performs self-attention (Q, K, V projections without bias, output projection with bias) and an MLP:

$$\text{FLOPs}_{\text{proc}}^{\text{fwd}} = 2D\Big[ 4W^2 N_h + \frac{WN_h^2}{16} + 2rW^2 N_h + 4WN_h \Big],$$

where the attention term uses a sparsity factor reflecting the graph structure.

The decoder projects hidden nodes back to grid nodes, mirroring the encoder but with reversed edge direction:

$$\text{FLOPs}_{\text{dec}}^{\text{fwd}} = 2\Big[ CWN_g + d_e WE_{\text{dec}} + 2W^2 N_h + 3W^2 N_g + WE_{\text{dec}} + 2rW^2 N_g + C_{\text{out}}WN_g \Big].$$

The total forward FLOPs are

$$\text{FLOPs}_{\text{AIFS}}^{\text{fwd}} = \text{FLOPs}_{\text{enc}}^{\text{fwd}} + \text{FLOPs}_{\text{proc}}^{\text{fwd}} + \text{FLOPs}_{\text{dec}}^{\text{fwd}};$$

training FLOPs are three times forward.

For AIFS, the dominant compute lies in the decoder due to the asymmetric graph structure: $N_g \gg N_h$ (e.g., 542,080 vs. 10,944), meaning decoder operations on grid nodes far exceed processor operations on hidden nodes. This contrasts with GraphCast, where mesh processing dominates. The encoder/decoder scale as $O(W^2)$ independent of depth, while the processor scales as $O(DW^2)$ but on the much smaller hidden graph.

Across models, the dominant FLOP contributions come from different operators. GraphCast is driven mainly by grid–mesh projections and repeated message passing on mesh edges and nodes. AIFS concentrates compute in its decoder, where hidden-to-grid message passing operates on the full grid resolution; the processor's transformer layers act on a reduced hidden graph and contribute less despite scaling with depth. Aurora is dominated by windowed self-attention and large MLPs in its Perceiver–Swin backbone. Pangu concentrates compute in its four-stage Swin blocks, with additional cost from down- and up-sampling projections at stage transitions. SFNO is characterized by the cost of spherical harmonic transforms and spectral multiplications inside Fourier blocks. Thus, each model's compute profile reflects its core inductive bias—graph structure, local attention, hierarchical features, or spectral representation.

For all models, the training FLOPs are approximately three times the forward FLOPs, reflecting the cost of both forward and backward passes. The factor of three arises because each operator must backpropagate both parameter gradients and input gradients, which requires two matrix multiplications of the same order as the forward pass plus elementwise derivatives. Throughout, $C$ denotes the effective number of input channels, treating prognostic variables, static fields, and forcings uniformly. Despite their architectural differences, all models share this common three-to-one training-to-forward compute ratio.

# D. Unified Validation Loss

For consistent evaluation, we implement the following alignment strategy for validation:

$$\mathcal{L}_{\text{MSE}} = \frac{1}{|D_{\text{batch}}|} \sum_{d_0 \in D_{\text{batch}}} \frac{1}{|G_{0.25°}|} \sum_{i \in G_{0.25°}} \sum_{j \in J} s_j \frac{w_j}{W} a_i \left( \hat{x}_{i,j}^{d_0+\tau} - x_{i,j}^{d_0+\tau} \right)^2, \tag{1}$$

The validation loss is defined in Equation (1), where:

- $D_{\text{batch}}$: set of forecast initial times in the mini-batch.

- $d_0 \in D_{\text{batch}}$: forecast initialization.

- $G_{0.25°}$: set of spatial grid cells at $0.25°$ resolution.

- $i \in G_{0.25°}$: spatial grid cell index.

- $J$: set of atmospheric variables and levels (e.g., $\{z1000, z850, \ldots, 2T, MSL\}$).

- $j \in J$: variable–level index.

- $W = \sum_{j \in J} w_j$: normalization constant.

- $w_j$: per-variable loss weight.

- $s_j$: per-variable inverse variance weight.

- $a_i$: normalized area of grid cell $i$ (mean area = 1).

- $\hat{x}_{i,j}^{d_0+\tau}$: model prediction for variable $j$ at grid cell $i$, lead time $\tau$, initialized from $d_0$.

- $x_{i,j}^{d_0+\tau}$: ground-truth ERA5 value for the same quantity.

The weights used in our unified validation loss are detailed in the tables below.

## D.1. Variable Definitions

Table 4 defines the variables used in the validation loss calculations, including their type, short name, and physical units.

*Table 4.* **Variable definitions.** The dash line represents the pressure level measured in hPa.

| Variable name | Type | Short name | Unit |
|---|---|---|---|
| 2m temperature | Surface | 2T | K |
| 10m u-component of wind | Surface | 10U | m/s |
| 10m v-component of wind | Surface | 10V | m/s |
| Mean sea level pressure | Surface | MSL | Pa |
| Temperature | Upper-air | t— | K |
| Specific humidity | Upper-air | q— | kg/kg |
| Geopotential | Upper-air | z— | $m^2/s^2$ |
| U-component of wind | Upper-air | u— | m/s |
| V-component of wind | Upper-air | v— | m/s |

## D.2. Loss Weights and Inverse Variance Weights of the Unified Validation Loss

Table 5 specifies the per-variable loss weight $w_j$ and the corresponding inverse variance weight $s_j$ for each specific atmospheric level or surface variable.

*Table 5.* **Per-variable loss weights and inverse variance weights.**

| Name | Loss weights $w_j$ | Inv. var. weights $s_j$ | Name | Loss weights $w_j$ | Inv. var. weights $s_j$ |
|---|---|---|---|---|---|
| 2T | 1.0 | $2.4101 \times 10^{-3}$ | z300 | $4.9793 \times 10^{-2}$ | $4.0072 \times 10^{-8}$ |
| 10U | 0.1 | $3.7612 \times 10^{-2}$ | z400 | $6.6390 \times 10^{-2}$ | $6.2134 \times 10^{-8}$ |
| 10V | 0.1 | $4.8160 \times 10^{-2}$ | z500 | $8.2988 \times 10^{-2}$ | $9.8700 \times 10^{-8}$ |
| MSL | 0.1 | $5.5049 \times 10^{-7}$ | z600 | $9.9585 \times 10^{-2}$ | $1.5930 \times 10^{-7}$ |
| t50 | $8.2988 \times 10^{-3}$ | $6.3288 \times 10^{-3}$ | z700 | $1.1618 \times 10^{-1}$ | $2.6508 \times 10^{-7}$ |
| t100 | $1.6598 \times 10^{-2}$ | $6.1954 \times 10^{-3}$ | z850 | $1.4108 \times 10^{-1}$ | $5.8404 \times 10^{-7}$ |
| t150 | $2.4896 \times 10^{-2}$ | $1.9150 \times 10^{-2}$ | z925 | $1.5353 \times 10^{-1}$ | $8.0029 \times 10^{-7}$ |
| t200 | $3.3195 \times 10^{-2}$ | $3.1566 \times 10^{-2}$ | z1000 | $1.6598 \times 10^{-1}$ | $9.0848 \times 10^{-7}$ |
| t250 | $4.1494 \times 10^{-2}$ | $1.4019 \times 10^{-2}$ | u50 | $8.2988 \times 10^{-3}$ | $5.7875 \times 10^{-3}$ |
| t300 | $4.9793 \times 10^{-2}$ | $7.9943 \times 10^{-3}$ | u100 | $1.6598 \times 10^{-2}$ | $5.5060 \times 10^{-3}$ |
| t400 | $6.6390 \times 10^{-2}$ | $5.6908 \times 10^{-3}$ | u150 | $2.4896 \times 10^{-2}$ | $3.8728 \times 10^{-3}$ |
| t500 | $8.2988 \times 10^{-2}$ | $5.4031 \times 10^{-3}$ | u200 | $3.3195 \times 10^{-2}$ | $3.2865 \times 10^{-3}$ |
| t600 | $9.9585 \times 10^{-2}$ | $5.2438 \times 10^{-3}$ | u250 | $4.1494 \times 10^{-2}$ | $3.1416 \times 10^{-3}$ |
| t700 | $1.1618 \times 10^{-1}$ | $4.9063 \times 10^{-3}$ | u300 | $4.9793 \times 10^{-2}$ | $3.4961 \times 10^{-3}$ |
| t850 | $1.4108 \times 10^{-1}$ | $4.6419 \times 10^{-3}$ | u400 | $6.6390 \times 10^{-2}$ | $5.1322 \times 10^{-3}$ |
| t925 | $1.5353 \times 10^{-1}$ | $4.2433 \times 10^{-3}$ | u500 | $8.2988 \times 10^{-2}$ | $7.5347 \times 10^{-3}$ |
| t1000 | $1.6598 \times 10^{-1}$ | $3.3497 \times 10^{-3}$ | u600 | $9.9585 \times 10^{-2}$ | $1.0323 \times 10^{-2}$ |
| q50 | $8.2988 \times 10^{-3}$ | $4.6903 \times 10^{13}$ | u700 | $1.1618 \times 10^{-1}$ | $1.3126 \times 10^{-2}$ |
| q100 | $1.6598 \times 10^{-2}$ | $2.7761 \times 10^{12}$ | u850 | $1.4108 \times 10^{-1}$ | $1.6636 \times 10^{-2}$ |
| q150 | $2.4896 \times 10^{-2}$ | $8.4827 \times 10^{10}$ | u925 | $1.5353 \times 10^{-1}$ | $1.8020 \times 10^{-2}$ |
| q200 | $3.3195 \times 10^{-2}$ | $2.0286 \times 10^{9}$ | u1000 | $1.6598 \times 10^{-1}$ | $3.0160 \times 10^{-2}$ |
| q250 | $4.1494 \times 10^{-2}$ | $1.8466 \times 10^{8}$ | v50 | $8.2988 \times 10^{-3}$ | $1.1710 \times 10^{-2}$ |
| q300 | $4.9793 \times 10^{-2}$ | $3.3694 \times 10^{7}$ | v100 | $1.6598 \times 10^{-2}$ | $1.2089 \times 10^{-2}$ |
| q400 | $6.6390 \times 10^{-2}$ | $3.4378 \times 10^{6}$ | v150 | $2.4896 \times 10^{-2}$ | $8.8805 \times 10^{-3}$ |
| q500 | $8.2988 \times 10^{-2}$ | $8.3635 \times 10^{5}$ | v200 | $3.3195 \times 10^{-2}$ | $7.0547 \times 10^{-3}$ |
| q600 | $9.9585 \times 10^{-2}$ | $3.2642 \times 10^{5}$ | v250 | $4.1494 \times 10^{-2}$ | $5.7463 \times 10^{-3}$ |
| q700 | $1.1618 \times 10^{-1}$ | $1.6113 \times 10^{5}$ | v300 | $4.9793 \times 10^{-2}$ | $5.6876 \times 10^{-3}$ |
| q850 | $1.4108 \times 10^{-1}$ | $6.0214 \times 10^{4}$ | v400 | $6.6390 \times 10^{-2}$ | $7.6670 \times 10^{-3}$ |
| q925 | $1.5353 \times 10^{-1}$ | $3.9040 \times 10^{4}$ | v500 | $8.2988 \times 10^{-2}$ | $1.1186 \times 10^{-2}$ |
| q1000 | $1.6598 \times 10^{-1}$ | $2.9064 \times 10^{4}$ | v600 | $9.9585 \times 10^{-2}$ | $1.5738 \times 10^{-2}$ |
| z50 | $8.2988 \times 10^{-3}$ | $4.8672 \times 10^{-8}$ | v700 | $1.1618 \times 10^{-1}$ | $2.0545 \times 10^{-2}$ |
| z100 | $1.6598 \times 10^{-2}$ | $3.9397 \times 10^{-8}$ | v850 | $1.4108 \times 10^{-1}$ | $2.5170 \times 10^{-2}$ |
| z150 | $2.4896 \times 10^{-2}$ | $3.1077 \times 10^{-8}$ | v925 | $1.5353 \times 10^{-1}$ | $2.4448 \times 10^{-2}$ |
| z200 | $3.3195 \times 10^{-2}$ | $3.0288 \times 10^{-8}$ | v1000 | $1.6598 \times 10^{-1}$ | $3.8239 \times 10^{-2}$ |
| z250 | $4.1494 \times 10^{-2}$ | $3.3499 \times 10^{-8}$ | | | |

## E. Variable-specific Loss Analysis

To evaluate how scaling trends manifest across different physical fields, we analyze loss and error separately for each variable. This approach allows us to assess whether improvements from increasing model capacity are uniform across variables or concentrated in specific subsets (e.g., large-scale dynamics versus moisture fields). Variables are not normalized, so comparisons reflect the inherent differences in scale and predictability across physical quantities rather than being adjusted to equal average contributions. This choice enhances the interpretability of per-variable scaling results, as they indicate which variables are intrinsically harder to predict and whether scaling benefits are distributed evenly across fields.

Following the weighting strategy used in GraphCast, we define:

$$\mathcal{L}_{\text{MSE}}(j) = \frac{1}{|D_{\text{batch}}|} \sum_{d_0 \in D_{\text{batch}}} \frac{1}{|G_{0.25°}|} \sum_{i \in G_{0.25°}} a_i \left( \hat{x}_{i,j}^{d_0} - x_{i,j}^{d_0} \right)^2, \tag{2}$$

where $j$ indexes the variable of interest and $a_i$ is the normalized grid-cell area. This procedure enables consistent reporting of per-variable RMSE, which are subsequently analyzed in Section 3.

## F. Probabilistic Analysis with Ensemble Forecasting

Weather prediction is inherently chaotic, with small perturbations in initial conditions potentially leading to widely diverging trajectories. To capture this uncertainty, we adopt an ensemble forecasting framework where multiple realizations are generated by perturbing the initial state or through stochastic model components. The resulting ensemble distribution provides a measure of forecast uncertainty and enables probabilistic skill evaluation.

A widely used metric in this context is the Continuous Ranked Probability Score (CRPS), which compares the forecast cumulative distribution function (CDF) $F$ with the observation $x$. Formally, CRPS is defined as

$$\text{CRPS}(F, x) = \int_{-\infty}^{\infty} [F(y) - \mathbf{1}\{y \geq x\}]^2 \, dy, \tag{3}$$

where $\mathbf{1}\{\cdot\}$ is the indicator function. For a finite ensemble $\{x_1, \ldots, x_N\}$ of size $N$, CRPS can be approximated by

$$\text{CRPS}(x_1, \ldots, x_N; x) = \frac{1}{N} \sum_{i=1}^{N} |x_i - x| - \frac{1}{2N^2} \sum_{i=1}^{N} \sum_{j=1}^{N} |x_i - x_j|. \tag{4}$$

CRPS rewards forecasts that are both sharp (low spread) and well-calibrated (ensemble mean close to the observation), reducing to the mean absolute error for deterministic single-member forecasts. In our experiments, we compute CRPS across variables for ensembles of size 10 and compare with larger ensembles to evaluate the trade-off between computational efficiency and probabilistic accuracy.

### F.1. Evaluation Metrics Comparison: MSE and CRPS

We compare MSE and CRPS across our scaling experiments. While CRPS curves show slight deviations from RMSE trends, they generally follow the same scaling relationships. CRPS computed with 10 ensemble members performs only 2–5% worse than with 50 members, suggesting diminishing returns from larger ensemble sizes in evaluation. For surface variables such as 2m temperature and 10m wind, the differences between ensemble sizes remain well below 0.05 K or 0.05 m/s, reinforcing the robustness of the scaling conclusions. Experiments with GraphCast (Figure 7) further show that once the MSE loss converges, CRPS begins to degrade slightly, likely reflecting mild overfitting of deterministic skill relative to probabilistic calibration. These results indicate that CRPS scaling mirrors RMSE scaling closely, and that smaller ensembles are sufficient for reliable evaluation in practice.

## G. Supplementary Tables for Model Specifications in Results

*Table 6.* **Model specifications for data-scaling analysis.** The first row for each model is transparent. For depth in tuple, it means the number of layers in different components of the Encoder. The Decoders have the same numbers in a reversed order. For simplicity, we omit them here.

| Model | Width | Depth | Size (M) |
|---|---|---|---|
| Aurora | 256 | (3, 5, 4) | 162.4 |
|  | 384 | (3, 5, 4) | 364.7 |
| GraphCast | 512 | 8 | 19.4 |
|  | 512 | 16 | 34.2 |
| SFNO | 384 | 8 | 289.4 |
|  | 512 | 8 | 514.5 |
| Pangu | 288 | (3,9) | 458.6 |
|  | 480 | (4,12) | 797.8 |
| AIFS | 256 | 16 | 37.5 |
|  | 512 | 16 | 80.6 |

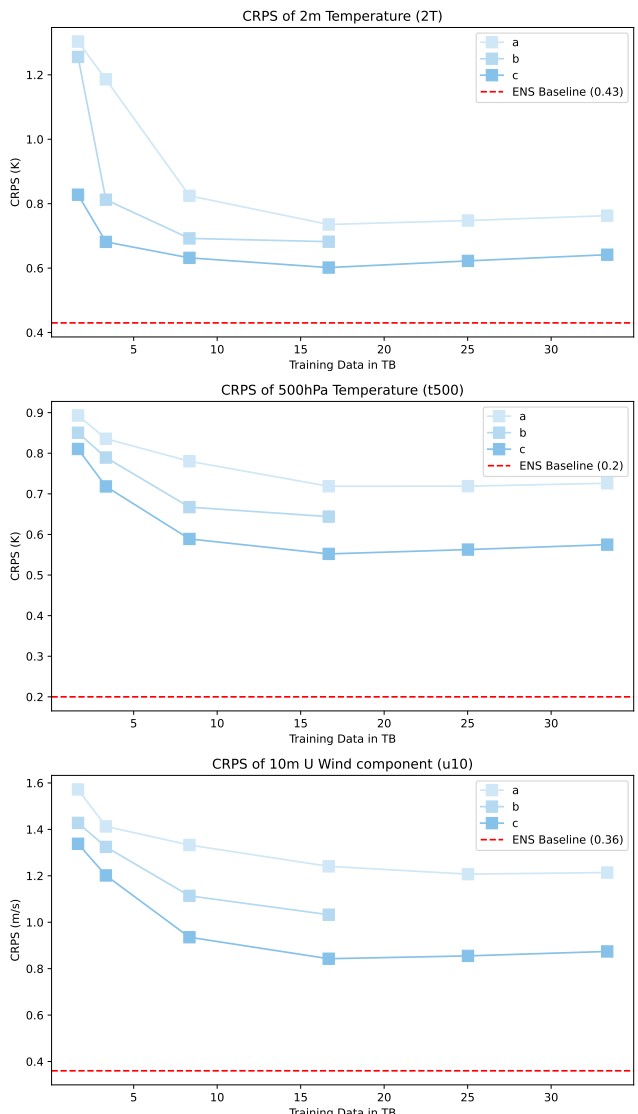

*Figure 7.* CRPS comparison for 2m temperature (2T), temperature at $500\,\text{hPa}$ (t500), and $10\,\text{m}$ $u$-velocity of wind (u10) using 10 ensemble members. Results with 10 ensembles are within 2–5% of those obtained with 50 members, indicating minimal loss of accuracy and supporting the efficiency of reduced ensemble sizes. Model specifications are provided in Table 8.

*Table 7.* **Model specifications for different shapes of same sizes.** For depth in tuple, it means the number of layers in different components of the Encoder. The Decoders have the same numbers in a reversed order. For simplicity, we omit them here.

| Model | Width | Depth | Size (M) |
|---|---|---|---|
| Aurora | 128 | (12, 20, 16) | 155.2 |
|  | 256 | (3, 5, 4) | 162.4 |
| GraphCast | 128 | 12 | 1.7 |
|  | 256 | 1 | 1.6 |
| SFNO | 256 | 8 | 128.6 |
|  | 512 | 2 | 129.1 |
| Pangu | 240 | (2, 6) | 290.0 |
|  | 288 | (2, 6) | 306.3 |
| AIFS | 128 | 16 | 26.7 |
|  | 256 | 4 | 28.0 |

*Table 8.* **Model shapes and sizes for CRPS experiments.**

| Model | Width | Depth | Size (M) |
|-------|-------|-------|----------|
| a | 128 | 4 | 0.8 |
| b | 256 | 8 | 4.9 |
| c | 512 | 16 | 34.2 |

*Table 9.* **Model specifications for variable-specific power law analysis.** For depth in tuple, it means the number of layers in different components of the Encoder. The Decoders have the same numbers in a reversed order. For simplicity, we omit them here.

| Model | Label | Width | Depth | Size (M) |
|-------|-------|-------|-------|----------|
| Aurora | Narrow | 128 | (3, 5, 4) | 40.8 |
| | Wide | 384 | (3, 5, 4) | 364.7 |
| GraphCast | Narrow | 128 | 16 | 2.1 |
| | Wide | 512 | 16 | 34.2 |
| Pangu | Narrow | 96 | (2, 6) | 258.8 |
| | Wide | 480 | (3, 9) | 599.4 |
| SFNO | Narrow | 128 | 8 | 32.2 |
| | Wide | 384 | 8 | 289.4 |
| AIFS | Narrow | 256 | 16 | 37.5 |
| | Wide | 512 | 8 | 55.4 |

*Table 10.* **Model specifications for all models presented in this work.** For depth in tuple, it means the number of layers in different components of the Encoder. The Decoders have the same numbers in a reversed order. For simplicity, we omit them here.

| Model | Width | Depth | Size (M) |
|---|---|---|---|
| Aurora | 64 | (3, 5, 4) | 10.3 |
| | 128 | (1, 1, 1) | 11.9 |
| | 64 | (6, 10, 8) | 19.9 |
| | 128 | (1, 2, 2) | 20.6 |
| | 128 | (3, 5, 4) | 40.8 |
| | 128 | (6, 10, 8) | 78.9 |
| | 384 | (1, 1, 1) | 105.0 |
| | 64 | (48, 80, 64) | 153.7 |
| | 128 | (12, 20, 16) | 155.2 |
| | 256 | (3, 5, 4) | 162.4 |
| | 192 | (6, 10, 8) | 177.2 |
| | 384 | (1, 2, 2) | 182.4 |
| | 384 | (1, 3, 2) | 204.1 |
| | 128 | (24, 40, 32) | 307.6 |
| | 384 | (3, 5, 4) | 364.7 |
| GraphCast | 128 | 1 | 0.4 |
| | 128 | 2 | 0.5 |
| | 128 | 4 | 0.8 |
| | 128 | 8 | 1.2 |
| | 256 | 1 | 1.6 |
| | 128 | 12 | 1.7 |
| | 256 | 2 | 2.1 |
| | 128 | 16 | 2.1 |
| | 256 | 4 | 3.0 |
| | 256 | 8 | 4.9 |
| | 512 | 1 | 6.6 |
| | 512 | 2 | 8.4 |
| | 256 | 16 | 8.6 |
| | 512 | 4 | 12.1 |
| | 512 | 8 | 19.4 |
| | 512 | 16 | 34.2 |

| Model | Width | Depth | Size (M) |
|---|---|---|---|
| Pangu | 96 | (2, 6) | 258.8 |
| | 192 | (2, 6) | 276.7 |
| | 240 | (2, 6) | 290.0 |
| | 288 | (2, 6) | 306.3 |
| | 288 | (3, 9) | 458.6 |
| | 480 | (3, 9) | 599.4 |
| | 480 | (4, 12) | 797.8 |
| SFNO | 128 | 1 | 4.1 |
| | 128 | 2 | 8.1 |
| | 128 | 4 | 16.1 |
| | 256 | 1 | 16.2 |
| | 128 | 8 | 32.2 |
| | 256 | 2 | 32.3 |
| | 384 | 1 | 36.5 |
| | 256 | 4 | 64.4 |
| | 512 | 1 | 64.9 |
| | 384 | 2 | 72.6 |
| | 256 | 8 | 128.6 |
| | 512 | 2 | 129.1 |
| | 384 | 4 | 144.9 |
| | 512 | 4 | 257.5 |
| | 384 | 8 | 289.4 |
| | 512 | 8 | 514.5 |
| AIFS | 64 | 4 | 23.4 |
| | 64 | 8 | 23.6 |
| | 64 | 16 | 24.0 |
| | 128 | 4 | 24.3 |
| | 128 | 8 | 25.1 |
| | 128 | 16 | 26.7 |
| | 256 | 4 | 28.0 |
| | 256 | 8 | 31.2 |
| | 256 | 16 | 37.5 |
| | 512 | 4 | 42.8 |
| | 512 | 8 | 55.4 |
| | 512 | 16 | 80.6 |
| | 1024 | 4 | 101.0 |
| | 1024 | 8 | 151.0 |
| | 1024 | 16 | 252.0 |

