# OpenReview forum: "Scaling Laws of Global Weather Models"
_ICML.cc/2026/Conference — ICML 2026 regular_

### Official Review · Reviewer_6WrQ · 2026-03-08

**Soundness:** 3
**Presentation:** 4
**Significance:** 2
**Originality:** 2
**Overall Recommendation:** 4
**Confidence:** 5

**Summary:**

Scaling laws for reanalysis variable prediction validation loss vs. model size, dataset size, and compute budget using five model architectures, Aurora, GraphCast, Pangu, SFNO, AIFS. Comparing dataset size scaling to model size scaling shows that weather models improve more quickly when scaling dataset size (note: authors do not state this carefully in current abstract). Tested models also show preference for width over depth scaling.

**Compliance With Llm Reviewing Policy:**

Affirmed.

**Final Justification:**

Bumped up score after rebuttals: The scope of this project is large, and merits broader visibility.

**Key Questions For Authors:**

* Line 012: “For resource allocation, our compute-optimal analysis indicates operational systems should prioritize smaller models trained for longer durations over big models with insufficient training [...] toward extending training steps [(Line 377)...]”: More precisely, reviewer believes that authors are claiming that training configurations should opt for smaller models trained on more total data (not just extending training duration or training steps, which could also be achieved with smaller batch sizes). Is that correct?
* RE: Getting to deeper insight:
   * Model inductive biases: Does the choice of graph structure, or mixing architecture (attention vs. messaging passing), etc. appear to contribute most to parameter or dataset scalability?
   * Can we interpret why inductive biases might scale as they do (e.g., in computer vision, convolutions provide a very parameter-efficient way to extract low-dimensional features, but it is unclear they are great for high-dimensional latents, where fully-connected transformer blocks tend to scale better with more data)?
   * Is it true that the underlying data generating function is expected close-to-linear, such that we would never expect depth to be very valuable? (see notes below) Or is there more to be investigated (e.g., maybe we are not appropriately parameterizing or normalizing models today)?
   * Ultimately, do the results suggest that there is room to further improve model architectures?
   * An example of a potentially deeper finding here: Authors state in the abstract: “weather forecasting models consistently favor increased width over depth. These findings suggest that future weather models should prioritize wider architectures and larger effective training datasets to maximize predictive performance.” The conclusion seems very first-order. Given that all tested model architectures appear to show the same width-over-depth scaling behavior, does this actually say something more about weather forecasting data, that it may be pretty linear in nature?
* RE: Methodology: Per weaknesses note, it seems like the training methods are ill-defined in the paper. Which optimizer(s) is used? How were learning rate schedules, batch sizes, other hyperparameters chosen? Is there any regularization? Are these decisions expected to be fair across models?
* How do scaling results differ for different variable-specific tasks? Authors suggest they do in Line 402, but omit any specifics: “These disparities stem both from model design biases—such as how different models represent spatial locality, anisotropy, and multi-scale interactions—and from the intrinsic difficulty of the variables themselves.“
* Do authors plan to contribute back their code repo changes or release their updated versions?

**Limitations:**

Adequate

**Strengths And Weaknesses:**

Strengths
* This paper covers a potentially very interesting breadth of models, model dimensionality, and training scales. Authors make an effort to control experiments for fair comparisons. Authors should be proud of the significant coverage and implementation effort in this work.
* The included results are already rich and worth marination and interpretation by researchers in forecasting and scaling laws. The experiments seem likely to be valid.
* The paper is well written and easy to follow.

Weaknesses
* At a high level, it seems like the current take-home insights from this paper have been noted in prior work… Though the paper studies a very interesting cross-section of model architectures on a broadly engaging domain (weather prediction), reviewer feels the discussion falls short of providing rich insight, especially given >400k GPU hours of testing here. Reviewer would have liked to learn more from such a broad study. Do these results suggest there is room to further improve model architectures? If so, how? Potential insights are requested in Key Questions.
* Although high-level training configuration is discussed, many details are omitted even from appendices. Unless the reader is quite familiar with the variety of tested model architectures (and related work), it is unclear whether the methodological decisions are fair to all models from prior work, and the results are likely irreproducible.


Including other notes reviewer collected while reading
* Important initial notes about scaling characteristics in Figures 2 and 3:
   * First, prior theoretical work suggests that the best possible scaling for a distance metric validation loss will have power-law curve fit exponent 0.5. There are a narrow set of conditions under which this scaling is possible, including that there must be a zero noise floor on the true data distribution (i.e., no stochasticity or irreducible error). For example, see “Four Types of Learning Curves”, Amari et al. 1993. However, it is important to note that as a validation loss, MSE might be (is?) a biased estimator for scaling characteristics, because it is the square of a linearized error metric, RMSE. The interpretation of power-law curve fits in Figures 2, 3, for example, might be optimistic, because squaring RMSE should increase the magnitude of power-law exponents by a factor of 2.
   * Still, the fact that results here for Aurora models show $\beta \approx 0.5$ (or $\beta_{\text{RMSE}} \approx 0.25$) in Figure 2 suggests that weather data (as collected) may have a well-behaved and smooth generating function with minimal noise. This alone is a very interesting result (assuming these tests were well-controlled - see further notes)!
   * Second, the fact that Figure 3 shows smaller magnitude power-law exponents for model size scaling than dataset size scaling (Figure 2) hints that model architectures are close to hitting “expressive enough” in their functional forms. Adding more dataset samples seems a better lever in modeling quality than model size (almost cubically so!). Again, assuming these tests were well-controlled, this would suggest that the underlying functional form of the true data generating function is simple and can maybe be represented with a compact or small model. A relevant paper discussing this trade-off is “Explaining Neural Scaling Laws”, Bahri et al., 2021: https://arxiv.org/abs/2102.06701
   * Relating Figures 2 and 3: These results are a bit difficult to interpret without some control on the compute budget (either FLOPs or wall-clock time) or compound interrelationship such as in Hoffmann et al. 2022 (Reviewer is looking forward to Section 3.3, just adding notes here…). In Figure 2, it would be good to note model size, since dataset size * model size is a reasonable proxy for compute budget. In Figure 3, there are points that appear to be missing from Figure 2? Compound scaling laws in both data and model size (e.g., Equation 2 in Hoffmann et al., 2022) provide clearer trade-offs in analysis of compute-efficiency. Nice: Authors do note a potentially confounding latent variable in their analysis in the next section: Model width vs. depth.
* Very cool: The results in Section 3.2 and Figure 4 show width is more important than depth. This result, yet again, hints that the underlying data generating function for weather is simple, as it may not require significant nonlinearity. (This is in contrast to many prior studies in computer vision and natural language, which suggest deeper models are more compute- and parameter-efficient). To make this study more holistic, reviewer would recommend sweeping width-to-depth ratios to assess whether depth and width should grow in some proportion. Example related work: “The Depth-to-Width Interplay in Self-Attention” Levine et al., 2020 https://arxiv.org/abs/2006.12467 . Line 225 (right): “Collectively, these findings imply that the default depths employed in these models may be greater than necessary, and suggest that the short-term dynamics of 6-hour weather prediction are approximated by linear models”
* Figure 2 shows two shades of points for each model architecture. What does each shade refer to? How is model size chosen here?
* Figure 3: Some model architectures show significant residuals from power-law fit (e.g., AIFS, Pangu). In general, for a given dataset size, we would expect that validation loss for a model architecture will saturate as we increase model size: There is only so much information available in a fixed-sized dataset. This is why most scaling laws papers will curve fit with a power-law+constant functional form (e.g., $\mathcal{L}(N) = \gamma N^{-\delta} + D$, where $D$ is an empirical constant). Perhaps this is what authors are referring to in Line 202 (right): “The results for Pangu reveal…”?
* Minor, Section 2.3: Authors precisely describe what they measure for validation loss, but only give reasoning for why a common metric is chosen (paragraph starting Line 146 (right)). It would be good to have a sense for why this particular metric was chosen. Is it fair to all models? Is it commonly used for all of them? Does its value correlate strongly with downstream tasks to which these models are commonly applied?
* Minor: “Conversely, in Transformer-based models, the patching mechanism reduces the effective $D$ by a factor of $p^2$.” Would recommend explicitly stating that $p$ is the patch size.
* Figure 5 results seem quite difficult to interpret…
   * Aurora data points are heavily intermixed and curve fits frequently cross. This makes it difficult to see any trend. The Pangu figure has flattened vertical axis making the data unreadable.
   * Line 313 (right): “as compute budgets increase, the optimal allocation strategy still favors increasing $D$ over $N$.” Since this note is already clear from Figures 2 and 3, could we deepen the analysis a bit further here? The ratio of $\beta$ to $\delta$ gives a sense for the relative speed of validation loss improvement along the dataset and parameter size dimensions. Does this vary by model architecture (very difficult to interpret from the plot), and if so, what does it say about the different architectures?
* Analogous to underdevelopment of the model scaling differences, reviewer feels the efficiency of algorithmic mapping to hardware (e.g., end of Section 4 and Table 2) feel underdeveloped.
* Reviewer believes that the peak FLOP/s numbers in Table 2 are incorrect. Recommend reviewing.

---

> ### Author Rebuttal · Authors · 2026-03-31
>
> We sincerely thank the reviewer for their incredibly thorough analysis, literature recommendations, and for recognizing the "significant coverage and implementation effort" of our study. We address your core questions regarding deep insights, methodology, and metrics below.
>
> Regarding your concern "At a high level ... in prior work", this paper systematically expands well-established scaling laws in language and vision domains to global weather models, which is non-trivial and strongly motivated by fundamental challenges including the chaotic nature and physical constraints of the atmosphere, and the unique spatial-temporal dependencies of atmospheric data. We expect to observe similarities of insights from prior work, which proves the generality and strengths of scaling laws.
>
> For "providing rich insight", we do provide suggestions for future model development. We suggest that future weather models should prioritize wider architectures and larger training datasets to maximize predictive performance. To test this, we further increased the width-to-depth ratio by setting depth=1 for GraphCast and SFNO and still observed robust performance. We also demonstrate the superior data-scaling of Swin-Transformer-based Aurora and the effectiveness of using attention in feature extraction such as Aurora instead of convolutional layers such as Pangu. Furthermore, we show that Aurora is more computationally efficient in practical training and deployment.
>
> For "methodological decisions" and "training methods", in general we followed the hyperparameter settings from the original papers to meet model preferences and tried different combinations for reasonably optimal training settings. The trickiest part is the learning rate and we used the largest fixed learning rate per model that enabled stable training for all model configurations. All models were trained using Distributed Data Parallel (DDP), utilizing Adam/AdamW optimizers. Details can be found in the table of Rebuttal to Reviewer nCRK and we will add them to the Appendix. For reproducibility, we provided the full training pipeline in the anonymous repo and are committed to publishing and updating the code afterwards.
>
> For your comment that "squaring RMSE ... by a factor of 2", we fully agree that the power-law exponents depend on the metric. We use MSE validation loss because it's widely used in weather models. We can replace all MSE loss with RMSE and there's no fundamental difference between MSE and RMSE affecting any log-log plot or the conclusions.
>
> For "Figure 2 shows two shades of points", we showed the detailed configurations in Table 3. The dark shade refers to the best (biggest) configuration for each model and the transparent shade shows representitive configuration at about half the size of the best. The points in Figure 2 and Figure 3 do not corresspond to each other - it would be too crowded to include 3-5 configurations per model in Figure 2, and vice versa. We performed the power-law plus constant fitting as shown in the repository folder figures/, which appears more accurate for Pangu and AIFS.
>
> We sincerely thank the reviewer for their keen observation that the peak FLOPs/s numbers appeared incorrect. We reviewed the calculation and found a bug in the code that swapped depth and width for GraphCast. The correct peak single-precision utilization for GraphCast on H100 GPUs is 1.03% (10.15 TFLOPs/s), not 0.017%. We followed the training pipeline in the public code base when provided and implemented the pipeline otherwise. Ultimately, the main contribution of this work is not optimizing training efficiency.
>
> For Key Question 1: Yes, we mean the dataset size. We will make it clear.
>
> For Key Question 2: 1) Yes, model design choices are the most important factor for scaling performance. Transformers appear to be better in dataset scalability while graph structure tends to achieve better performance under similar parameter counts. 2) We compare the performance of Aurora and Pangu to show the efficiency of attention-based feature extraction compared to convolutional layers. (Line 91 Right) 3) At least in the 6-hour lead time forecast setting, the encoded weather data appears to be approximated by linear models. However, it requires further investigation for longer lead times. 4) Indeed there is room for improvements. Please refer to paragraph three. 5) Yes, this conclusion is valid across model architectures and configurations, which provides evidence for the underlying nature of the dataset itself.
>
> Key Question 3 and 5 are already answered.
>
> For Key Question 4: Specifically, we observe that variables like 2-meter temperature (2t) are intrinsically easier to predict and therefore exhibit much better scaling behavior. We will explicitly include these concrete examples in the revised text to ground our statement.
>
> Thank you once again for your time, expertise, and thoughtful engagement, which have undoubtedly strengthened the clarity and impact of our paper.

---

> > ### Author Rebuttal · Reviewer_6WrQ · 2026-04-03
> >
> > Reviewer thanks the authors for their review response. Generally, this paper provides broad insight on scaling laws for weather forecasting.
> >
> > This reviewer is part-way between weak reject and weak accept: Although the authors describe a few observations about model architecture choices, those observations are almost purely descriptive (“what”) rather than explanatory (“why”). A couple examples:
> > * Ablating width and depth and then stating "width is more important than depth" is a contextualized description of the relative value of these hyperparameters, but it doesn't give the reader any understanding if there might be something wrong with the models or training procedure (Aside: Nearly all scaling laws papers that the reviewer has reviewed have incorrect hyperparameter choices for scaling, and those choices easily change the conclusions of the studies). Similarly, comparing graph-based and transformer models is a great way to gather deeper insight, but authors have only stated the top-level result that transformer scales better with FLOPs, not as well with parameters. As pointed out in the original review, things like training parameterization or normalization could be hindering model scalability by blocking models from making use of more depth. Understanding whether parameterization or normalization are problems, or whether graph-based models are inherently poor inductive biases for weather modeling are more impactful insights to the community.
> > * Related to the comparison of model architectures, the current paper doesn’t give quantitative comparison of the optimal scaling characteristics for the different model architectures, which could give more explanatory comparison of the models. For instance, what is the optimal trade-off of dataset size vs. model size? Does this give some intuition why the loss for Pangu models predominantly determined by dataset size? Is there something wrong with that architecture?

---

> > > ### Author Response · Authors · 2026-04-08
> > >
> > > We sincerely thank the reviewer for acknowledging our rebuttal and for highlighting that our paper provides "broad insight on scaling laws for weather forecasting." We also agree that understanding the precise reasons behind model behaviors is the ultimate goal for this field.
> > >
> > > Regarding the concern that the observed scaling behavior stems from suboptimal training procedures, we observe a consistent performance advantage for wider configurations across five distinct models under the same dataset and validation loss. Furthermore, for SFNO and AIFS, our experiments were conducted using official training frameworks (please refer to https://github.com/NVIDIA/makani/blob/main/makani/train.py and https://github.com/ecmwf/anemoi-core/blob/main/training/src/anemoi/training/train/train.py).
> > > We achieved stable training dynamics and consistently decreasing validation loss curves across all models, suggesting that our results reflect structural scaling properties rather than suboptimal training. We have detailed our hyperparameter ablation studies in our reply to Reviewer nCRK.
> > >
> > > Regarding insights into comparison between Graph-based and Transformer models, we do provide analysis beyond top-level data- and parameter-scaling. We noted that GraphCast relies on GNN-based message passing with Multi-Layer Perceptrons, which avoids the quadratic costs of attention and achieve higher parameter efficiency whereas Transformer-based Aurora better leverages GPU Tensor Cores for massive dense matrix multiplications of attention. Though AIFS has GNNs in the encoder and decoder, it utilizes attention on the grid points in the backbone and exhibits a mixed feature from Graph-based and Transformer models. While Aurora (Transformer) has the highest FLOPs/step, its high GPU utilization (37.2%, more than 10x higher than other models) makes it the most efficient model in practical deployment (please refer to Sec.4).
> > >
> > > Regarding the concern that training parameterization or normalization might hinder models from utilizing greater depth, we find no evidence that optimization bottlenecks (like normalization or parameterization) hindered depth scaling. In fact, there’s consistent performance gain from depth scaling, yet there’s even more advantage from width scaling. For the training procedure, all models used robust, standard configurations (AdamW/Adam+L2 regularization, and native LayerNorm) that ensured stability. Therefore, we hypothesize that the advantage of width over depth is driven by the properties of weather data. In a 6-hour lead time setting, encoded atmospheric dynamics at short lead times appear well-approximated by linear transformations. To test this, we pushed the width-to-depth ratio to the limit by setting depth=1 for GraphCast and SFNO. Even with a single message-passing step or SFNO block, we observed robust predictive performance. This suggests that once the atmospheric state is adequately embedded—a process that benefits from width—deep sequential processing is less critical for short-term forecasting.
> > >
> > > Regarding Pangu’s observed behavior, we identify two potential reasons. (1) the absence of an official training pipeline, necessitating a PyTorch re-implementation that may introduce implementation variance, and (2) a discrepancy between its native MAE training objective and our unified MSE/RMSE validation, which likely leads to suboptimal convergence in the initial training stages.
> > >
> > > Finally, regarding the optimal trade-off between dataset and model size, we are currently constrained by GPU memory (limiting width to 1024) and total compute budget. As noted in the Pangu and Aurora papers, the models haven’t yet reached full convergence given massive budgets (100 epochs and 5k GPU hours, respectively).
> > >
> > > We sincerely thank you for your time and expertise, which have undoubtedly strengthened the clarity of our paper. We hope that our deeper explanations of the underlying mechanisms have addressed your reservations.

---

### Official Review · Reviewer_RXwn · 2026-03-11

**Soundness:** 3
**Presentation:** 4
**Significance:** 3
**Originality:** 3
**Overall Recommendation:** 5
**Confidence:** 4

**Summary:**

This paper is the first to systematically explore Scaling Laws in the field of data-driven global weather forecasting. Its core findings include:
1. Width is superior to depth, indicating that weather forecasting relies more on high representational power (width) than on deep nonlinear logical reasoning (depth).
2. Computational allocation strategies. Under a fixed computational budget, extending training time (increasing data volume) is more effective in reducing prediction errors than blindly increasing model size.
3. A series of experiments demonstrate significant differences between models and between variables.

**Compliance With Llm Reviewing Policy:**

Affirmed.

**Final Justification:**

I believe this is a very valuable work, and the authors have thoroughly addressed my current questions during the rebuttal process. I haven't found any significant issues or weakness. Overall, I think this paper meets the acceptance criteria.

**Key Questions For Authors:**

See 'Strengths And Weaknesses'.

**Limitations:**

yes

**Strengths And Weaknesses:**

In this work, the authors conducted extensive benchmarking tests on five mainstream architectures (GraphCast, AIFS, Aurora, Pangu, SFNO) using the same ERA5 dataset and evaluation criteria, through massive experiments (430,000 GPU hours). **Overall, I believe this is a good paper, requiring only minor revisions**. Its strengths are:

1. It answers a key question in the AI4Earth field. To my knowledge, some works have attempted similar preliminary scaling experiments, but these have generally been limited to small-scale experiments. This work should be the first comprehensive scaling law experiment. This is of reference value to the AI4Earth community, and even the broader AI4Science community.
2. Interesting and insightful conclusions. The authors propose a counterintuitive conclusion: the scaling behavior of meteorological models differs fundamentally from that of large language models (LLMs): under a fixed parameter budget, increasing the width of a meteorological model yields significantly greater benefits than increasing its depth. This provides a good direction for designing the next generation of data-driven meteorological models.
3. Hardware Utilization Analysis. The article goes beyond theoretical parameter analysis and delves into hardware utilization. For example, it points out that while GraphCast has the lowest validation loss with the same number of parameters, its computational utilization on an H100 GPU is only 0.017% due to its message passing mechanism (compared to Aurora's 37.2%). This perspective, combining theory with practical engineering deployment, is very comprehensive.

There are some minor issues about this work, which requires further clarifications:

4. Contextualizing Hardware Utilization. When discussing computational efficiency, the paper notes that GraphCast's peak single-precision utilization on H100 GPUs is only 0.017%, whereas Aurora reaches 37.2%. Given that the appendix mentions rewriting GraphCast's JAX training pipeline for multi-node training, it would be helpful to add a brief discussion in Section 4. Specifically, the authors could clarify to what extent this extremely low utilization is an inherent limitation of the GNN message-passing mechanism versus a bottleneck in the current engineering implementation.

5. Transparency of Loss Weights. Appendix C details the unified validation loss formula (1), which includes the inverse variance weight $s_j$ and the variable loss weight $w_j$. While Section 2.3 roughly explains the weighting logic for upper-air and surface variables (like 2m temperature and 10m wind speed), it is recommended to provide a complete table in the appendix. Listing the exact weight values for all evaluated variables would greatly help the community achieve perfect open-source reproducibility.

---

> ### Author Rebuttal · Authors · 2026-03-31
>
> We thank the reviewer for the "Accept" recommendation and for the detailed appreciation of our study’s scale (430K GPU hours) and our findings regarding model width vs. depth. We address the two points for clarification below.
>
> We thank the reviewer for highlighting the hardware utilization analysis. We reviewed the calculation and found a bug in the code that swapped depth and width for GraphCast. The correct peak single-precision utilization for GraphCast on H100 GPUs is 1.03% (10.15 TFLOPs/s), not 0.017%. While this corrected figure is significantly higher than previously stated, it remains low compared to Aurora (37.2%). This gap is largely an inherent limitation of the GNN message-passing mechanism, which involves irregular memory access and high communication overhead that scales poorly on high-throughput Tensor Core architectures like the H100.
>
> We appreciate the suggestion to improve reproducibility. We will move this information from Section 2.3 into a dedicated table in the Appendix as requested. The weights used in our unified validation loss are listed in the tables below.
>
> We thank the reviewer again for their careful review, and all clarifications above will be updated in the final revision.
>
> | **Variable Name** | **Type**  | **Short Name** | **Unit** |
> | ----------------------- | --------- | -------------- | -------- |
> | 2m Temperature| Surface   | 2T   | K  |
> | 10m U-component of wind | Surface   | 10U  | m/s|
> | 10m V-component of wind | Surface   | 10V  | m/s|
> | Mean Sea Level Pressure | Surface   | MSL  | Pa |
> | Temperature   | Upper-air | t--- | K  |
> | Specific Humidity | Upper-air | q--- | kg/kg    |
> | Geopotential  | Upper-air | z--- | m²/s²    |
> | U-component of wind     | Upper-air | u--- | m/s|
> | V-component of wind     | Upper-air | v--- | m/s|
>
> | **Short Name** | Variable loss weight $w_j$ | **Inverse Variance Weight** $s_j$ |
> | -------------- | -------------------------- | --------------------------------- |
> | 2T   | 1.0       | 2.4101e-3|
> | 10U  | 0.1       | 3.7612e-2|
> | 10V  | 0.1       | 4.8160e-2|
> | MSL  | 0.1       | 5.5049e-7|
> | t50  | 8.2988e-3 | 6.3288e-3|
> | t100 | 1.6598e-2 | 6.1954e-3|
> | t150 | 2.4896e-2 | 1.9150e-2|
> | t200 | 3.3195e-2 | 3.1566e-2|
> | t250 | 4.1494e-2 | 1.4019e-2|
> | t300 | 4.9793e-2 | 7.9943e-3|
> | t400 | 6.6390e-2 | 5.6908e-3|
> | t500 | 8.2988e-2 | 5.4031e-3|
> | t600 | 9.9585e-2 | 5.2438e-3|
> | t700 | 1.1618e-1 | 4.9063e-3|
> | t850 | 1.4108e-1 | 4.6419e-3|
> | t925 | 1.5353e-1 | 4.2433e-3|
> | t1000| 1.6598e-1 | 3.3497e-3|
> | q50  | 8.2988e-3 | 4.6903e+13       |
> | q100 | 1.6598e-2 | 2.7761e+12       |
> | q150 | 2.4896e-2 | 8.4827e+10       |
> | q200 | 3.3195e-2 | 2.0286e+9|
> | q250 | 4.1494e-2 | 1.8466e+8|
> | q300 | 4.9793e-2 | 3.3694e+7|
> | q400 | 6.6390e-2 | 3.4378e+6|
> | q500 | 8.2988e-2 | 8.3635e+5|
> | q600 | 9.9585e-2 | 3.2642e+5|
> | q700 | 1.1618e-1 | 1.6113e+5|
> | q850 | 1.4108e-1 | 6.0214e+4|
> | q925 | 1.5353e-1 | 3.9040e+4|
> | q1000| 1.6598e-1 | 2.9064e+4|
> | z50  | 8.2988e-3 | 4.8672e-8|
> | z100 | 1.6598e-2 | 3.9397e-8|
> | z150 | 2.4896e-2 | 3.1077e-8|
> | z200 | 3.3195e-2 | 3.0288e-8|
> | z250 | 4.1494e-2 | 3.3499e-8|
> | z300 | 4.9793e-2 | 4.0072e-8|
> | z400 | 6.6390e-2 | 6.2134e-8|
> | z500 | 8.2988e-2 | 9.8700e-8|
> | z600 | 9.9585e-2 | 1.5930e-7|
> | z700 | 1.1618e-1 | 2.6508e-7|
> | z850 | 1.4108e-1 | 5.8404e-7|
> | z925 | 1.5353e-1 | 8.0029e-7|
> | z1000| 1.6598e-1 | 9.0848e-7|
> | u50  | 8.2988e-3 | 5.7875e-3|
> | u100 | 1.6598e-2 | 5.5060e-3|
> | u150 | 2.4896e-2 | 3.8728e-3|
> | u200 | 3.3195e-2 | 3.2865e-3|
> | u250 | 4.1494e-2 | 3.1416e-3|
> | u300 | 4.9793e-2 | 3.4961e-3|
> | u400 | 6.6390e-2 | 5.1322e-3|
> | u500 | 8.2988e-2 | 7.5347e-3|
> | u600 | 9.9585e-2 | 1.0323e-2|
> | u700 | 1.1618e-1 | 1.3126e-2|
> | u850 | 1.4108e-1 | 1.6636e-2|
> | u925 | 1.5353e-1 | 1.8020e-2|
> | u1000| 1.6598e-1 | 3.0160e-2|
> | v50  | 8.2988e-3 | 1.1710e-2|
> | v100 | 1.6598e-2 | 1.2089e-2|
> | v150 | 2.4896e-2 | 8.8805e-3|
> | v200 | 3.3195e-2 | 7.0547e-3|
> | v250 | 4.1494e-2 | 5.7463e-3|
> | v300 | 4.9793e-2 | 5.6876e-3|
> | v400 | 6.6390e-2 | 7.6670e-3|
> | v500 | 8.2988e-2 | 1.1186e-2|
> | v600 | 9.9585e-2 | 1.5738e-2|
> | v700 | 1.1618e-1 | 2.0545e-2|
> | v850 | 1.4108e-1 | 2.5170e-2|
> | v925 | 1.5353e-1 | 2.4448e-2|
> | v1000| 1.6598e-1 | 3.8239e-2|

---

> > ### Author Rebuttal · Reviewer_RXwn · 2026-04-02
> >
> > Thanks for the author's concise and efficient response.
> >
> > The authors have completely answered all the questions I raised during the review process, especially the corrections and clarifications regarding the GraphCast hardware utilization data, and the supplementary complete variable loss weight table. These explanations were very clear and helpful.
> >
> > Based on the above response and the contributions of the paper itself, I have decided to stand by my initial decision and recommend accepting this work.

---

> > > ### Author Response · Authors · 2026-04-07
> > >
> > > We are grateful for the reviewer’s positive feedback and the recommendation for acceptance. We are glad that our clarifications and the supplementary data were helpful, and we thank the reviewer for the insightful comments that improved the quality of our work.

---

### Official Review · Reviewer_nCRK · 2026-03-13

**Soundness:** 2
**Presentation:** 3
**Significance:** 3
**Originality:** 3
**Overall Recommendation:** 5
**Confidence:** 4

**Summary:**

This paper presents an empirical study of scaling laws across different types of weather models (Aurora, GraphCast, SFNO, Pangu, AIFS) on the ERA5 dataset, measuring how validation loss relates to model size, dataset size, compute budget. The authors find that Aurora exhibits the strongest data-scaling efficiency, GraphCast demonstrates the best parameter efficiency, and all models consistently favour increased width over depth. The results also show that under fixed compute allocations, greater gains can be achieved when increasing the training duration, in contrast to increasing the number of model parameters.

**Compliance With Llm Reviewing Policy:**

Affirmed.

**Final Justification:**

Following the rebuttal and additional details on the hyperparameters, I have raised my score to fully recommend acceptance of the paper.

**Key Questions For Authors:**

- What optimiser was used, what parallelisation strategies (data/model parallel training) was used?

**Limitations:**

see weaknesses

**Strengths And Weaknesses:**

Strengths
- The first study in the literature to consider a broad range of weather model architectures to derive scaling laws. The unified training setup also indicates a major engineering effort.
- The finding and explanation of shallower but wider networks being preferred is convincing and is an interesting contrast to language model scaling laws
- Not only validation loss, but also individual variables (such as wind and temperature) is considered in the analysis, though the findings are not conclusive

Weaknesses
- While the study features a significant investment of compute resources (0.43M GPU hours), the reporting of the training and hyperparameter setup (Section 2.2) is surprisingly shallow. Since model performance is known to be highly sensitive to at least learning rate and batch size, their omission casts doubt on the findings, particularly given that different model architectures are compared, each of which may have distinct hyperparameter preferences. While a full hyperparameter ablation for each model is infeasible at this scale, at least some targeted ablations are crucial to ensure each model's training setup is reasonably optimal.
- GraphCast was likely optimised for TPU training, yet the analysis focuses solely on GPU efficiency, potentially disadvantaging it in a comparison it was not designed for.

---

> ### Author Rebuttal · Authors · 2026-03-31
>
> We thank the reviewer for their constructive feedback and for recognising our work as a "major engineering effort" that "advances the sub-area of AI."
>
> For your question “What optimiser was used, what parallelisation strategies (data/model parallel training) was used”, the specific configurations are summarized in the table below and will be added to the Appendix. All models were trained using Distributed Data Parallel (DDP), utilizing Adam/AdamW optimisers. Regarding the concern about "distinct hyperparameter preferences," we conducted preliminary sweeps to ensure stability and optimality for each model. In general, we followed the hyperparameter settings in the original papers to meet model preferences and tried different combinations for reasonably optimal training settings. For example, the learning rate was tuned per model, and we used the largest learning rate that enabled stable training for all model configurations.
>
> | **Model**     | **Optimiser** | **Fixed Learning Rate** | **Betas**    | **Parallelization** | **Batch Size** |
> | ------------- | ------------- | ----------------------- | ------------ | ------------------- | -------------- |
> | **Aurora**    | AdamW | $5 \times 10^{-4}$      | (0.9, 0.999) | DDP| 16   |
> | **GraphCast** | Adam| $1 \times 10^{-4}$      | (0.9, 0.999) | DDP| 16   |
> | **Pangu**     | Adam| $5 \times 10^{-4}$      | (0.9, 0.999) | DDP| 4    |
> | **SFNO**      | AdamW | $1 \times 10^{-4}$      | (0.9, 0.95)  | DDP| 16   |
> | **AIFS**      | AdamW | $1 \times 10^{-4}$      | (0.9, 0.95)  | DDP| 16   |
>
> For your concern that “GraphCast was likely optimised for TPU training", we acknowledge that hardware can affect training efficiency. However, it’s not an apples-to-apples comparison if we test GraphCast on different hardware to the H100 GPU. There is also an implementation of GraphCast training pipeline on A100 GPUs published on PMLR (*Fixing the Double Penalty in Data-Driven Weather Forecasting Through a Modified Spherical Harmonic Loss Function*, Subich et al., 2025).
>
> Furthermore, we calculate the utilisation on TPU and find that the performance gap does not reflect significant degradation on GPU. Based on the training duration (\~4 weeks), the FLOPs/step (\~48.8 TFLOPs/step), and the schedule (300k steps with one roll-out step and 11k steps of increasing roll-out steps from 2 to 12, totaling 371.5k steps), we have the performance of 7.49 TFLOPs/s, which is 2.73% utilisation of 275 TFLOPs/s bfloat16 peak on the Google TPU v4. Regarding the extremely low utilisation of GraphCast on H100, we reviewed the calculation and found a bug in the code that swapped depth and width. The corrected performance of GraphCast is 10.15 TFLOPs/s, which is 1.03% utilisation of 32-bit peak performance.
>
> We hope these clarifications, along with the corrected utilisation metrics and detailed configurations added to the revised manuscript, fully address your concerns regarding our experimental fairness and hardware evaluation.

---

> > ### Author Rebuttal · Reviewer_nCRK · 2026-04-02
> >
> > I thank the authors for their clarifications. My concerns regarding the TPU/GPU efficiency of the GraphCast model have been addressed, and the table with the learning rate and batch size configurations is a welcome addition. However, regarding the hyperparameter choices, open questions remain. As mentioned in my original review, some targeted ablations are crucial. The authors mention that these were apparently performed ("we conducted preliminary sweeps to ensure stability and optimality for each model"), but do not offer concrete details (such as sweep intervals, training durations, or the range of configurations tested) on what these sweeps consisted of.

---

> > > ### Author Response · Authors · 2026-04-08
> > >
> > > We thank the reviewer for the positive assessment and for acknowledging that our earlier clarifications addressed the concerns regarding TPU/GPU efficiency and training configurations. We also appreciate the request for more concrete details on the hyperparameter sweeps, and we agree that making these explicit improves transparency and reproducibility. We will add these configurations to the appendix of the camera-ready version.
> > >
> > > For Aurora, we swept learning rates of 3e-6, 1.2e-5, 3e-5, 1e-4, and 5e-4, as well as batch sizes of 16 and 32 (at 5e-4), evaluated over one 6-hour rollout step for more than 3000 steps. By following the original paper for settings such as the constant learning rate schedule after warmup and patch size of 2 for Swin Transformer, 5e-4 provided a balance between the rate of loss reduction and training stability.
> > >
> > > For AIFS, following the original scaling rules, we used a local learning rate of 6.25e-6, yielding an effective learning rate of 1e-4 for a batch size of 16. We additionally evaluated Maximal Update Parametrization (μP) [1] on the largest model (width 1024) with an adjusted rate of 0.707e-4 over more than 3000 training steps (corresponding to over 40TB of dataset size), but observed consistently degraded validation loss curves. Furthermore, the 1e-4 effective rate caused loss spikes on the largest model, confirming we were operating near the maximum stable boundary.
> > >
> > > For GraphCast, while the original paper used 1e-3, we swept 1e-4, 3e-4, and 1e-3 for our batch size of 16. Rates of 1e-3 and 3e-4 were unstable and diverged early (between 1000-2000 steps), so we scaled down to 1e-4, which stabilized the loss curve.
> > >
> > > For Pangu, we tested the identical learning rate of 5e-4 from the original paper for a batch size of 4. Under this configuration, the validation loss exhibits a steady downward trend throughout the training process, making further hyperparameter ablations unnecessary.
> > >
> > > For SFNO, we swept a range of learning rates of 1e-5, 1e-4, and the original paper's 1e-3 for a batch size of 16. The original 1e-3 proved unstable for our setup, while 1e-4 provided the best downward trend without instability.
> > >
> > > Finally, we note that exhaustive hyperparameter grid searches are computationally prohibitive at this scale. Moreover, we observed that training these weather models is sensitive to the learning rate. Too big learning rates resulted in validation loss divergence, even when the training loss remained stable. Furthermore, tuning heuristics for large language models (eg. μP) do not directly transfer, as fitting continuous weather data differs from predicting discrete text tokens. Compounding this, each weather model exhibits different hyperparameter preferences, requiring tailored configurations based on the original papers. Nevertheless, our empirical tuning achieved stable training dynamics with a consistently decreasing validation loss across the evaluated models.
> > >
> > > We thank you again for your time, expertise, and the constructive dialogue regarding our paper. We hope our clarifications have addressed your remaining questions.
> > >
> > > [1] Yang, Ge, et al. "Tuning large neural networks via zero-shot hyperparameter transfer." Advances in Neural Information Processing Systems 34 (2021): 17084-17097.

---

### Official Review · Reviewer_cfNT · 2026-03-17

**Soundness:** 4
**Presentation:** 4
**Significance:** 4
**Originality:** 3
**Overall Recommendation:** 4
**Confidence:** 1

**Summary:**

As the title suggests, this paper is about studying the scaling laws of global weather models.

**Compliance With Llm Reviewing Policy:**

Affirmed.

**Final Justification:**

Thanks to the authors for the response! I maintain my initial evaluation of a weak accept with very low confidence.

**Key Questions For Authors:**

NA

**Limitations:**

Societal impact is discussed. I did not find an explicit section on limitations near the societal impact section, but I could be missing something.

**Strengths And Weaknesses:**

I would like to be transparent that I feel somewhat conflicted about this paper, though I appreciate several of its strengths.

In terms of limitations, my understanding is that the work does not introduce fundamentally new methods, and the analytical approach may not be highly novel. One possible perspective is that the paper builds on existing models and established analysis, extending them through large-scale experimentation. That said, there may well be meaningful novelty in the specific domain and in the insights obtained. As this is not my primary area of expertise, I do not feel fully qualified to assess those contributions in depth.

Overall, I find myself leaning slightly toward acceptance, albeit with low confidence. The problem domain appears relatively new and important, and I acknowledge my own limited familiarity with it. In that context, studies like this can provide valuable contributions. I also want to commend the authors for the systematic and careful execution of their experiments. The scale of computation involved (430K GPU hours) is particularly impressive and goes well beyond what is typically feasible in academic settings, which further enhances the potential value of the findings.

---

> ### Author Rebuttal · Authors · 2026-03-31
>
> We thank the reviewer for their transparency and for rating our Soundness, Presentation, and Significance as "Excellent." We appreciate the recognition of our systematic experimental execution (430K GPU hours).
>
> For novelty and methodology, we acknowledge that while scaling laws are an established framework in NLP, their application to global weather modeling is non-trivial. As pointed out in Sec.1 Line 78, “applying existing scaling laws in the language and vision domains to weather models faces several fundamental challenges”, including the chaotic nature and physical constraints of the atmosphere, and the unique spatial-temporal dependencies of atmospheric data. To the best of our knowledge, this paper provides the first systematic study of scaling laws in this domain, which we believe offers significant value to researchers in the AI4Earth community.
>
> For the limitations, we will add an explicit section or paragraph. In the paper, we have already discussed the limitation of scaling laws that computational efficiency is excluded from scaling laws analysis, yet they critically determine model deployment efficiency in practice. In addition, we also discussed the memory constraints and training budget constraints that limit us from exploring the full compute-optimal frontier in Section 3.3.
>
> Given the reviewer’s positive assessment of our study’s significance and execution, we hope these clarifications address the concerns regarding novelty and limitations.

---

> > ### Author Rebuttal · Reviewer_cfNT · 2026-04-03
> >
> > Thanks to the authors for their response! I have updated the scores. I believe I initially meant for a weak accept with very low confidence, but I might have entered the wrong rating.
> >
> > I maintained my original opinion, slightly leaning toward acceptance with very low confidence.

---

> > > ### Author Response · Authors · 2026-04-07
> > >
> > > We sincerely thank the reviewer for the updated positive score. We are glad to have addressed your initial concerns and we appreciate your time and effort in evaluating our manuscript.

---

### Decision · Program_Chairs · 2026-04-30

**Decision:**

Accept (regular)

**Comment:**

This paper presents the first systematic study of scaling laws for data-driven global weather forecasting, training five leading architectures (Aurora, GraphCast, Pangu, SFNO, AIFS) on ERA5 under a unified setup at roughly 430k GPU-hours and fitting power laws in model size $N$, dataset size $D$, and compute $C$. The headline findings are: Aurora exhibits the strongest data-scaling exponent ($\beta \approx 0.51$); under fixed compute, allocating to more training data dominates allocating to more parameters; and across all five architectures, wider-shallower configurations outperform deeper-narrower ones at fixed parameter count (in contrast to language LLM). Reviewers agreed that the controlled scale of the comparison and the width-over-depth finding are valuable contributions to both the AI4Earth community and the broader scaling-law literature, and the scores are uniformly above positive. I recommend acceptance.

For the camera-ready, the authors should honor the following commitments made during the rebuttal:

- **Training-setup transparency.** The optimizer, learning rate, batch size, and parallelism details that were absent from Section 2.2 should be moved into the appendix as a complete table, together with the per-model learning-rate sweeps the authors provided in their follow-up to Reviewer nCRK (Aurora, AIFS, GraphCast, Pangu, SFNO). Reviewer 6WrQ flagged reproducibility as otherwise at risk, and the authors' detailed sweep description should be preserved in full rather than paraphrased.
- **GraphCast utilization correction.** The GraphCast H100 utilization in Table 2 and the associated text must be corrected: the authors acknowledged a code bug that swapped depth and width, and the correct figure is 10.15 TFLOPs/s (1.03% utilization), not 0.017%.
- **Abstract wording.** The phrase "longer training durations" should be rewritten to say more total training data, per the authors' agreement with Reviewer 6WrQ, a small but meaningful distinction for scaling-law claims.
- **$\mu$P ablation brought into the paper.** The $\mu$P ablation that the authors reported in their follow-up to Reviewer nCRK should be incorporated into the paper. A clean negative $\mu$P result from a large controlled scaling study is a valuable contribution to the broader ML community's ongoing conversation about how $\mu$P.